# Automatic detection of 39 fundus diseases and conditions in retinal photographs using deep neural networks

Ling-Ping Cen [1,6], Jie Ji [2,3,4,6], Jian-Wei Lin[1], Si-Tong Ju[1], Hong-Jie Lin[1], Tai-Ping Li[1], Yun Wang[1], Jian-Feng Yang[1], Yu-Fen Liu[1], Shaoying Tan[1], Li Tan[1], Dongjie Li[1], Yifan Wang[1], Dezhi Zheng[1], Yongqun Xiong[1], Hanfu Wu[1], Jingjing Jiang[1], Zhenggen Wu[1], Dingguo Huang[1], Tingkun Shi[1], Binyao Chen[1], Jianling Yang[1], Xiaoling Zhang[1], Li Luo[1], Chukai Huang[1], Guihua Zhang[1], Yuqiang Huang[1], Tsz Kin Ng[1,3,5], Haoyu Chen[1], Weiqi Chen [1], Chi Pui Pang[1,5] & Mingzhi Zhang [1✉]

Retinal fundus diseases can lead to irreversible visual impairment without timely diagnoses and appropriate treatments. Single disease-based deep learning algorithms had been developed for the detection of diabetic retinopathy, age-related macular degeneration, and glaucoma. Here, we developed a deep learning platform (DLP) capable of detecting multiple common referable fundus diseases and conditions (39 classes) by using 249,620 fundus images marked with 275,543 labels from heterogenous sources. Our DLP achieved a frequency-weighted average F1 score of 0.923, sensitivity of 0.978, specificity of 0.996 and area under the receiver operating characteristic curve (AUC) of 0.9984 for multi-label classification in the primary test dataset and reached the average level of retina specialists. External multihospital test, public data test and tele-reading application also showed high efficiency for multiple retinal diseases and conditions detection. These results indicate that our DLP can be applied for retinal fundus disease triage, especially in remote areas around the world.

[1] Joint Shantou International Eye Centre of Shantou University and The Chinese University of Hong Kong, Shantou, Guangdong, China. [2] Network & Information Centre, Shantou University, Shantou, Guangdong, China. [3] Shantou University Medical College, Shantou, Guangdong, China. [4] XuanShi Med Tech (Shanghai) Company Limited, Shanghai, China. [5] Department of Ophthalmology and Visual Sciences, The Chinese University of Hong Kong, Shatin, Hong Kong. [6] These authors contributed equally: Ling-Ping Cen, Jie Ji. ✉email: zmz@jsiec.org

Millions of people in the world are affected by ocular fundus diseases such as diabetic retinopathy (DR)[1], age-related macular degeneration (AMD)[2], retinal vein occlusion (RVO)[3], retinal artery occlusion (RAO)[4], glaucoma[5], retinal detachment (RD), and fundus tumors[6]. Among them, DR, AMD, and glaucoma are the most common cause of vision impairment in most populations. Without accurate diagnoses and timely appropriate treatment[7,8], these fundus diseases can lead to irreversible blurred vision, metamorphopsia, visual field defect, or even blindness. However, in rural and remote areas, especially in developing countries, where there is insufficiency in ophthalmic service and a shortage of ophthalmologists, early detection and timely referral for treatment may not be available. Notably, fundus photography that provides basic detection of these diseases is available and affordable in most parts of the world[9]. A fundus photo can be handled by non-professorial personnel and delivered online to major ophthalmic institutions for follow-up. Artificial intelligence (AI) is able to provide delivery capability.

AI is an established but still rapidly advancing technology, especially in computer-aided diagnosis of human diseases[10]. It has been effectively applied in the detection of Alzheimer's disease[11], intracranial diseases[12], arrhythmia[13], skin cancer[14], lung cancer[4], mesothelioma[15], lymph node metastases of breast cancer[16], and colorectal cancer[17]. In retinal diseases, deep learning algorithms for AI-assisted diagnoses have been applied to screen for DR[18,19], AMD[20,21], retinopathy of prematurity[22,23], glaucoma[24], and papilledema[25]. These AI-assisted diagnosis systems mostly focus on the detection of a single retinal disease. In clinical practice, retinal disease screening of single-disease diagnostic algorithm, for example for DR, would not recognize other fundus diseases such as AMD, glaucoma, RVO, and RAO. In real-life, especially in remote areas lacking specialized ophthalmologists, the capability to efficiently detect various types of fundus diseases is needed. A multi-disease detecting system using fundus images should be developed to avoid missed diagnoses and consequently delayed treatment.

In this study, we have developed a multi-disease automatic detection platform by applying convolutional neural networks (CNNs) constructed in a customized two-step strategy that can classify 39 types of common fundus diseases and conditions based on color fundus images (Supplementary Table 1). We have established a deep learning platform (DLP) that was trained, validated, and tested with 249,620 fundus images collected from multiethnic data sets of multiple hospitals in different parts of China, a data set from the United States, and four public data sets. It is capable to predict the probability of each disease and display heatmaps providing deep learning explainability in real-time.

## Results

**Data characteristics and system architecture.** Fundus images from 7 diverse data sources were collected for deep learning algorithm development and validation (Table 1 and Methods: "Data sets and labeling" section). Among them, the primary data set for training, validation, and test was collected from the Picture Archiving and Communication Systems (PACS) at Joint Shantou International Eye Centre (JSIEC) in China, the Lifeline Express Diabetic Retinopathy Screening Systems (LEDRS) in China, and the Eye Picture Archive Communication System (EyePACS) in the United States. (see Methods: "Data sets and labeling" section). Primary data set acquisition and processing flow are shown in Supplementary Figs. 1 and 2. Besides the primary test data sets, the DLP was further evaluated by external multihospital test, public data test, and tele-reading application.

In total, 249,620 images marked with 275,543 labels were collected for algorithm training, validation and tests. Patient demographics and image characteristics are summarized in Table 1. Inter-grader agreements and images labeled as unclassifiable in each data set were analyzed (Supplementary Tables 2 and 3). The training data set of totally 129,264 images was collected from JSIEC ($n = 74,683$), LEDRS ($n = 27,463$) and EyePACS ($n = 27,118$). The validation and tests data sets contained another 120,356 images that had not been "seen" by the algorithm during the training process. They consisted of five parts: (1) Primary validation data set ($n = 22,800$) collected from JSIEC ($n = 13,247$), LEDRS ($n = 4787$) and EyePACS ($n = 4766$). (2) Primary test data set ($n = 27,611$) collected from JSIEC 2018 ($n = 14,502$), LEDRS 2018 ($n = 7052$) and EyePACS ($n = 6057$). (3) External multihospital test data sets ($n = 60,445$) collected from three hospitals, one in Fujian in southeastern China ($n = 39,671$), Tibet in western China ($n = 14,826$) and Xinjiang in northwestern China ($n = 5948$). (4) Four publicly available data sets ($n = 3438$): Messidor-2 ($n = 1748$), Indian Diabetic Retinopathy Image Data set (IDRID) ($n = 516$), Pathological Myopia (PALM) ($n = 374$), and Retinal Fundus Glaucoma challenge (REFUGE) ($n = 800$). (5) Tele-reading ($n = 6062$) was conducted in seven hospitals located in different parts of China. The hetero-ethnic characteristics of the test data sets enable effective assessment of generalization capability of our DLP in disease classification.

We developed a two-level hierarchical system for the classification of the 39 types of diseases and conditions (Supplementary Table 1) with three groups of CNNs and Mask-RCNN (Supplementary Table 4 and Supplementary Fig. 3) using data sets of fundus images as described above. The system was then deployed into the production environment as a platform for internal testing. The whole data flow and simplified architecture of the DLP are shown in Supplementary Fig. 1 and Supplementary Fig. 4. Technical details of algorithms and implementation are explicated in the Methods: "Architecture of the DLP" section and Supplementary Methods: "Algorithm development and the DLP deployment" section.

Image data distribution and results in various classes for each data set are shown in Supplementary Tables 5–10. Positive samples of a class were obtained by summing up its false negatives (FN) and true positives (TP) accordingly. The data sets were extremely imbalanced and the labels were sparse in some bigclasses, as indicated by the imbalance ratio of negative samples vs positive samples (Supplementary Fig. 5).

**Performance in primary test.** Appearance of the preprocessed images and heatmaps of examples of diseases and conditions are shown in Fig. 1. Results of the primary test data set showed that in the 30-bigclass detection evaluation, we achieved a referable frequency-weighted average F1 score of 0.923, sensitivity of 0.978, specificity of 0.996, and AUC of 0.9984 (Table 2). We termed major diseases or conditions as "bigclass" for convenient classification and statistical analysis. Details of true positive (TP), false positive (FP), true negative (TN), and false negative (FN) of each bigclass are shown in Supplementary Table 7. Range of $F_1$ scores for every bigclass was 0.766–0.983. The highest $F_1$ scores were achieved in bigclass with obvious features such as RVO (0.983), maculopathy (0.965), silicon oil in eye (0.964), and laser spots (0.967). The lowest $F_1$ scores, on the contrary, were obtained in bigclasses with ambiguous features such as posterior serous/exudative RD (0.829), optic nerve degeneration (0.852), severe hypertensive retinopathy (0.829), chorioretinal atrophy/coloboma (0.861), and preretinal hemorrhage (0.766). Sensitivity and specificity for detection of referable bigclasses were above 0.942 and 0.979, respectively. ROCs were generated to evaluate the ability of the DLP to detect every bigclass (Fig. 2a). The DLP achieved an

**Table 1 Summary of data sets.**

| Data sets | Age[a], mean (SD), y | Men[a], no. (%) | No. | | | | | |
|---|---|---|---|---|---|---|---|---|
| | | | Labels | Images | Subjects | Referable (%) | | |
| | | | | | | Labels | Images | Subjects |
| **Training** | | | | | | | | |
| JSIEC | 51.7 (18.2) | 18,088 (50.0) | 87,594 | 74,683 | 36,156 | 74,940 (85.6) | 62,029 (83.1) | 31,223 (86.4) |
| LEDRS | 61.6 (10.4) | 5485 (44.8) | 29,851 | 27,463 | 12,236 | 15,682 (52.5) | 13,294 (48.4) | 7215 (59.0) |
| EYEPACS | N/A | N/A | 27,743 | 27,118 | 19,751 | 9194 (33.1) | 8569 (31.6) | 7333 (37.1) |
| Total, training | | | 145,188 | 129,264 | 68,143 | 99,816 (68.7) | 83,892 (64.9) | 45,771 (67.2) |
| **Validation** | | | | | | | | |
| JSIEC | 52.0 (18.2) | 3179 (49.7) | 15,451 | 13,247 | 6392 | 13,193 (85.4) | 10,989 (83.0) | 5519 (86.3) |
| LEDRS | 61.3 (10.4) | 918 (43.3) | 5229 | 4787 | 2118 | 2812 (53.8) | 2370 (49.5) | 1284 (60.6) |
| EYEPACS | N/A | N/A | 4857 | 4766 | 3475 | 1518 (31.3) | 1427 (29.9) | 1229 (35.4) |
| Total, validation | | | 25,537 | 22,800 | 11,985 | 17,523 (68.6) | 14,786 (64.9) | 8032 (67.0) |
| Total, training, and validation | | | 170,725 | 152,064 | 80,128 | 117,339 (68.7) | 98,678 (64.9) | 53,803 (67.1) |
| **Test** | | | | | | | | |
| JSIEC | 51.1 (19.4) | 4005 (49.2) | 16,851 | 14,502 | 8146 | 13,778 (81.8) | 11,429 (78.8) | 6901 (84.7) |
| LEDRS | 64.8 (10.3) | 1,313 (39.0) | 7455 | 7052 | 3364 | 4239 (56.9) | 3836 (54.4) | 2080 (61.8) |
| EYEPACS | N/A | N/A | 6906 | 6057 | 5262 | 5264 (76.2) | 4415 (72.9) | 3800 (72.2) |
| Total, test | | | 31,212 | 27,611 | 16,772 | 23,281 (74.6) | 19,680 (71.3) | 12,781 (76.2) |
| Total, training, validation, and test | | | 201,937 | 179,675 | 96,900 | 140,620 (69.6) | 118,358 (65.9) | 66,584 (68.7) |
| **Multihospital tests** | | | | | | | | |
| Fujian | N/A | N/A | 41,410 | 39,671 | 19,810 | 14,799 (35.7) | 13,060 (32.9) | 7200 (36.3) |
| Tibet[b] | 55.4 (18.1) | 3955 (50.3) | 15,981 | 14,826 | 7864 | 10,234 (64.0) | 9079 (61.2) | 5205 (66.2) |
| Xinjiang[b] | N/A | N/A | 6263 | 5948 | 1669 | 3923 (62.6) | 3608 (60.7) | 1016 (60.9) |
| Total, multihospital test | | | 63,654 | 60,445 | 29,343 | 28,956 (45.5) | 25,747 (42.6) | 13,421 (45.7) |
| Public tests[c] | N/A | N/A | 3438 | 3438 | N/A | 1019 (29.6) | 1019 (29.6) | N/A |
| Tele-reading categorized | N/A | N/A | 6514 | 6062 | 3251 | 3944 (60.5) | 3492 (57.6) | 1882 (57.9) |
| Total, training, validation, test, multihospital tests, public test, and tele-reading categorized | | | 275,543 | 249,620 | 129,494 | 174,539 (63.3) | 148,616 (59.5) | 81,887 (63.2) |

[a]Age and gender information cannot be obtained are marked as "N/A".
[b]97.1% subjects of Tibet data set are Tibetan, 13.3% subjects of Xinjiang data set are Uygur.
[c]Public tests include four public data sets in single-disease setting: Messidor-2, Indian Diabetic Retinopathy Image Data set (IDRID), Pathological Myopia (PALM), and Retinal Fundus Glaucoma challenge (REFUGE).

area under the ROC greater than 0.99 for all referable bigclasses. Subset accuracy, which evaluated the fraction of correctly detected cases with identical label set of prediction and ground-truth, was 87.98% for the whole primary test data set.

Multi-class classification of subclasses in bigclass 0, 1, 2, 5, 10, 15, and 29 was further evaluated after the 30-bigclass detection evaluation. $F_1$ score above 0.8 was achieved for all subclasses except DR1, which was nonreferable DR, with a $F_1$ score of 0.479. The sensitivity and specificity obtained were greater than 0.8 and 0.9, respectively, for all subclasses except DR1. Details of the other results for subclass analysis in the primary test data set are shown in Supplementary Table 11 and Supplementary Fig. 6.

Results of primary training and primary validation data sets for bigclasses are provided in Supplementary Tables 5 and 6.

**Generalization tests on heterogeneous image data sets.** To verify the generalization abilities for the detection of multiple diseases and conditions, the DLP was further tested with three heterogeneous data sets collected from hospitals with patients of different ethnicities and publicly available data sets (Table 1). Results for $F_1$ score, sensitivity, specificity, and AUC of 30 bigclasses of the multihospital data set are shown in Table 2. We achieved a referable frequency-weighted average $F_1$ score of 0.920, sensitivity of 0.971, specificity of 0.998, and AUC of 0.999. Range of $F_1$ scores for every bigclass was 0.652–0.984. Sensitivity and specificity for detection of all referable bigclasses were above 0.855 and 0.982, respectively. The DLP achieved an area under the ROC of greater than 0.99 for all referable bigclasses (Fig. 2b).

A higher subset accuracy (92.62%) for the whole multihospital data set than that of the primary test data set was obtained. This may due to the different data distributions of much higher percentage of nonreferable images in the multihospital test data set.

After testing with hetero-ethic data sets in China, the generalization capabilities of our DLP to detect different diseases and conditions were evaluated with 4 public test data sets in the single-disease setting, messidor-2, IDRID, PALM, and REFUGE (Supplementary Table 9). For detecting referable DR, we achieved a $F_1$ score of 0.944, sensitivity of 0.906, specificity of 0.996 and AUC of 0.9861 in messidor-2. Performance was weaker in IDRID with $F_1$ score of 0.875, sensitivity of 0.824, specificity of 0.902 and AUC of 0.9431. We reviewed the misjudged cases and found the presence of stains on dirty lens looking like hemorrhage spots as the main cause of false positive results. For pathological myopia, higher performance in PALM was achieved with a $F_1$ score of 0.974, sensitivity of 0.958, specificity of 0.988, and AUC of 0.9931. Performance was moderate for optic nerve degeneration (possible glaucoma) with a $F_1$ score of 0.651 (0.674), sensitivity of 0.850 (0.813), specificity of 0.915 (0.933), and AUC of 0.9397 when compared to the top 12 contestant teams in the REFUGE Challenge[26]. Labels of all images in the FEFUGE data set were initially confirmed by multiple examinations including intraocular pressure (IOP), optical coherence tomography (OCT), and visual field. In early-stage glaucoma, almost no noticeable change could be detected by fundus images through OCT could show retinal nerve fiber layer thinning. These cases were missed by the DLP which was developed based on fundus images only. Without

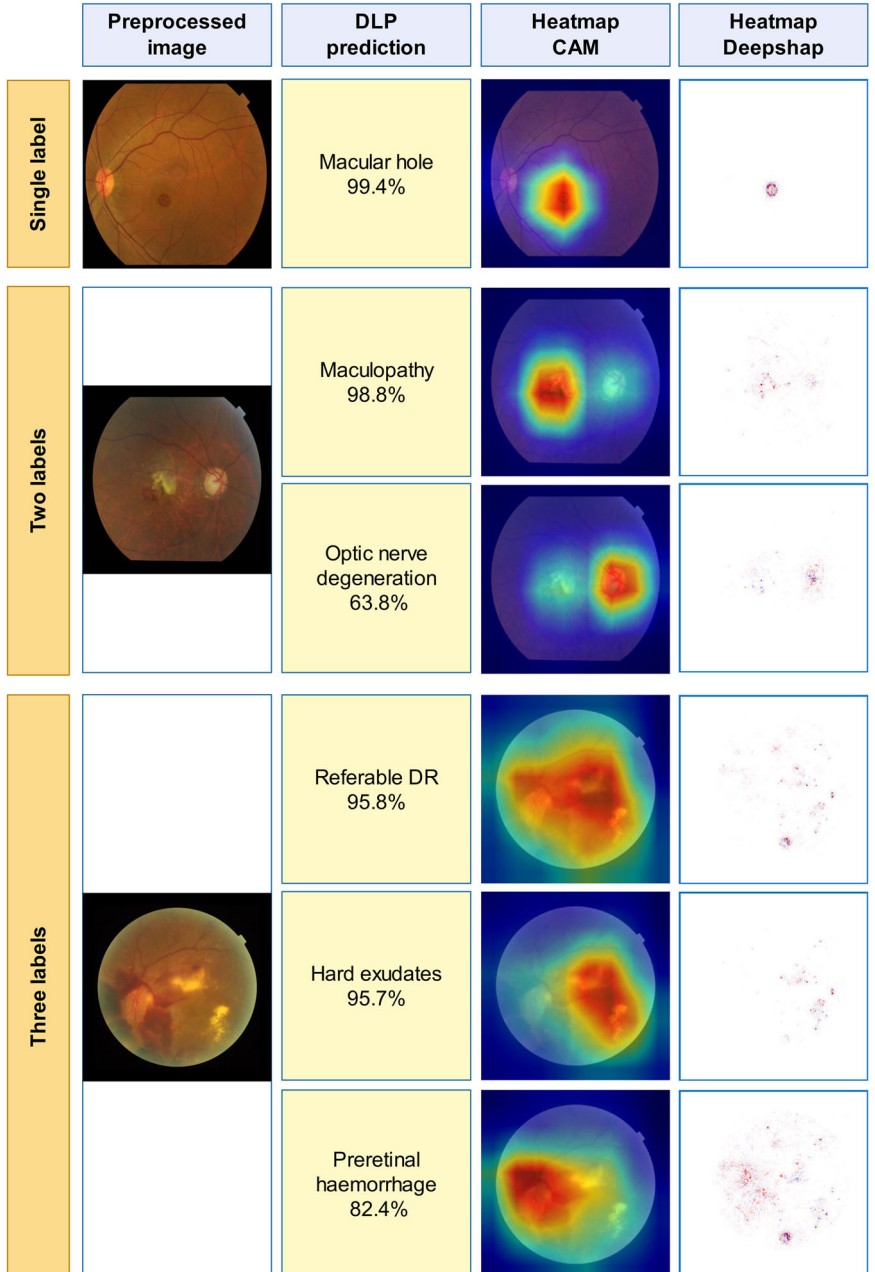

**Fig. 1 Examples of preprocessed images and heatmaps.** Typical images with one label, two labels, and three labels were selected and shown for heatmaps. First column: typical preprocessed image of selected images with a resolution of 299 × 299 pixels. Second column: their predictions. Third and fourth columns: the heatmaps (CAM and Deepsharp) indicating important regions with typical features of diseases discovered by DLP for predictions.

specific optimization to the data sets of single diseases for competitions, the overall performance of our multi-label DLP with single-disease data sets in different distributions was acceptable. These indicate good generalization capabilities in our DLP for detecting fundus diseases in heterogeneous images.

**Achieving expert performance in comparative test**. To further validate the diagnostic competence of DLP, a comparative test was conducted between the DLP and five retinal specialists with more than 10 years of clinical experiences. A comparative test data set consisting of 922 images was collected from PACS JSIEC with patient information (711 images collected from February to December 2018) and selected from the external test data sets (without patient information) of three hospitals in different

localities of China: Fujian ($n = 85$), Tibet ($n = 74$), and Xinjiang ($n = 52$). These images had not been "seen" by the DLP and included various challenging diseases and conditions, which were arranged by the retina expert panel. The majority decision served as the reference standard for classification. The five retinal specialists were requested to complete the whole test independently as the AI without patient information by selecting different class labels for each image, followed by an additional test containing the same JSIEC image data along with patient information. The final results were the average of the five specialists for every bigclass.

Detailed results of the comparative test are summarized in Table 3 and Supplementary Table 12. AUC analysis and ROC of some bigclasses are shown in Fig. 3. For the whole test without patient information, the retina specialists achieved a referable

**Table 2 Performance of DLP for classification of bigclasses in primary test and multihospital tests data sets.**

| Diseases/conditions | ID | Primary test data set (n = 27,611) | | | | | Multihospital tests data set (n = 60,445) | | | | |
|---|---|---|---|---|---|---|---|---|---|---|---|
| | | $F_1$ | Sensitivity | Specificity | AUC (95% CI) | Subset accuracy[b], % | $F_1$ | Sensitivity | Specificity | AUC (95% CI) | Subset accuracy, % |
| Non-referable | 0 | 0.939 | 0.898 | 0.995 | 0.9888 (0.9878–0.9898) | | 0.965 | 0.942 | 0.987 | 0.9765 (0.9754–0.9775) | |
| Referable DR | 1 | 0.938 | 0.969 | 0.979 | 0.9936 (0.9928–0.9945) | | 0.945 | 0.981 | 0.997 | 0.9987 (0.9984–0.9990) | |
| RVO | 2 | 0.983 | 0.978 | 1.000 | 0.9985 (0.9965–1.0000) | | 0.970 | 0.957 | 1.000 | 0.9995 (0.9992–0.9998) | |
| RAO | 3 | 0.938 | 0.950 | 1.000 | 1.0000 (0.9999–1.0000) | | 0.933 | 0.933 | 1.000 | 0.9999 (0.9997–1.0000) | |
| Rhegmatogenous RD | 4 | 0.950 | 0.988 | 0.998 | 0.9997 (0.9994–0.9999) | | 0.924 | 0.993 | 0.999 | 0.9998 (0.9998–0.9999) | |
| Posterior serous/exudative RD | 5 | 0.829 | 0.990 | 0.997 | 0.9994 (0.9985–1.0000) | | 0.794 | 0.994 | 0.999 | 0.9996 (0.9995–0.9998) | |
| Maculopathy | 6 | 0.965 | 0.976 | 0.998 | 0.9991 (0.9989–0.9994) | | 0.944 | 0.945 | 0.999 | 0.9990 (0.9988–0.9992) | |
| ERM | 7 | 0.871 | 0.958 | 0.991 | 0.9972 (0.9964–0.9980) | | 0.836 | 0.938 | 0.996 | 0.9976 (0.9969–0.9982) | |
| MH | 8 | 0.924 | 0.984 | 0.999 | 0.9997 (0.9994–1.0000) | | 0.933 | 0.942 | 1.000 | 0.9996 (0.9991–1.0000) | |
| Pathological myopia | 9 | 0.943 | 0.997 | 0.995 | 0.9997 (0.9996–0.9998) | | 0.949 | 0.990 | 0.996 | 0.9994 (0.9993–0.9995) | |
| Optic nerve degeneration | 10 | 0.852 | 0.992 | 0.987 | 0.9964 (0.9958–0.9969) | | 0.843 | 0.958 | 0.991 | 0.9972 (0.9967–0.9977) | |
| Severe hypertensive retinopathy | 11 | 0.829 | 1.000 | 0.999 | 0.9993 (0.9990–0.9996) | | 0.897 | 0.855 | 1.000 | 0.9998 (0.9989–0.9998) | |
| Disc swelling and elevation | 12 | 0.897 | 0.997 | 0.997 | 0.9997 (0.9995–0.9998) | | 0.873 | 0.980 | 0.998 | 0.9992 (0.9989–0.9995) | |
| Dragged disc | 13 | 0.892 | 1.000 | 1.000 | 0.9999 (0.9998–1.0000) | | 0.857 | 1.000 | 1.000 | 0.9999 (0.9999–1.0000) | |
| Congenital disc abnormality | 14 | 0.909 | 1.000 | 1.000 | 0.9999 (0.9997–1.0000) | | 0.884 | 1.000 | 1.000 | 0.9998 (0.9995–1.0000) | |
| Pigmentary degeneration | 15 | 0.898 | 1.000 | 0.998 | 0.9999 (0.9999–1.0000) | | 0.918 | 0.990 | 0.998 | 0.9995 (0.9992–0.9999) | |
| Peripheral retinal degeneration and break | 16 | 0.870 | 1.000 | 0.995 | 0.9997 (0.9996–0.9998) | | 0.652 | 1.000 | 0.998 | 0.9995 (0.9993–0.9997) | |
| Myelinated nerve fiber | 17 | 0.950 | 0.950 | 1.000 | 0.9956 (0.9873–1.0000) | | 0.952 | 0.926 | 1.000 | 0.9998 (0.9996–0.9999) | |
| Vitreous particles | 18 | 0.899 | 1.000 | 0.999 | 1.0000 (0.9999–1.0000) | | 0.936 | 0.996 | 1.000 | 1.0000 (0.9999–1.0000) | |
| Fundus neoplasm | 19 | 0.900 | 1.000 | 1.000 | 0.9999 (0.9998–1.0000) | | 0.848 | 0.929 | 1.000 | 0.9999 (0.9998–1.0000) | |
| Hard exudates | 20 | 0.925 | 0.999 | 0.995 | 0.9989 (0.9985–0.9993) | | 0.922 | 0.995 | 0.998 | 0.9996 (0.9994–0.9997) | |
| Yellow-white spots/flecks | 21 | 0.915 | 0.976 | 0.982 | 0.9927 (0.9910–0.9944) | | 0.909 | 0.959 | 0.992 | 0.9969 (0.9965–0.9974) | |
| Cotton-wool spots | 22 | 0.938 | 0.984 | 0.997 | 0.9987 (0.9981–0.9993) | | 0.758 | 0.933 | 0.999 | 0.9988 (0.9983–0.9993) | |
| Vessel tortuosity | 23 | 0.906 | 0.981 | 0.998 | 0.9996 (0.9994–0.9997) | | 0.896 | 0.942 | 0.998 | 0.9985 (0.9982–0.9988) | |
| Chorioretinal atrophy/coloboma | 24 | 0.861 | 0.942 | 0.998 | 0.9962 (0.9921–1.0000) | | 0.900 | 0.933 | 0.997 | 0.9976 (0.9968–0.9983) | |
| Preretinal hemorrhage | 25 | 0.766 | 0.997 | 0.993 | 0.9985 (0.9979–0.9991) | | 0.766 | 0.986 | 0.997 | 0.9990 (0.9983–0.9998) | |
| Fibrosis | 26 | 0.892 | 0.994 | 0.997 | 0.9992 (0.9988–0.9995) | | 0.820 | 0.986 | 0.998 | 0.9995 (0.9993–0.9997) | |
| Laser spots | 27 | 0.967 | 0.955 | 1.000 | 0.9996 (0.9994–0.9997) | | 0.984 | 0.979 | 1.000 | 0.9999 (0.9999–1.0000) | |
| Silicon oil in eye | 28 | 0.964 | 0.992 | 0.999 | 0.9991 (0.9974–1.0000) | | 0.892 | 0.947 | 0.999 | 0.9969 (0.9928–1.0000) | |
| Blur fundus | 29 | 0.940 | 0.978 | 0.979 | 0.9961 (0.9953–0.9970) | | 0.943 | 0.986 | 0.982 | 0.9964 (0.9959–0.9968) | |
| Referable[a], frequency-weighted average | | 0.923 | 0.978 | 0.996 | 0.9984 | | 0.920 | 0.971 | 0.998 | 0.9990 | |
| Total | | | | | | 87.98 | | | | | 92.62 |

[a]Referable refers to bigclass 1–29.
[b]Subset accuracy measures the scale of samples having identical labels between DLP prediction and the ground-truth labels.

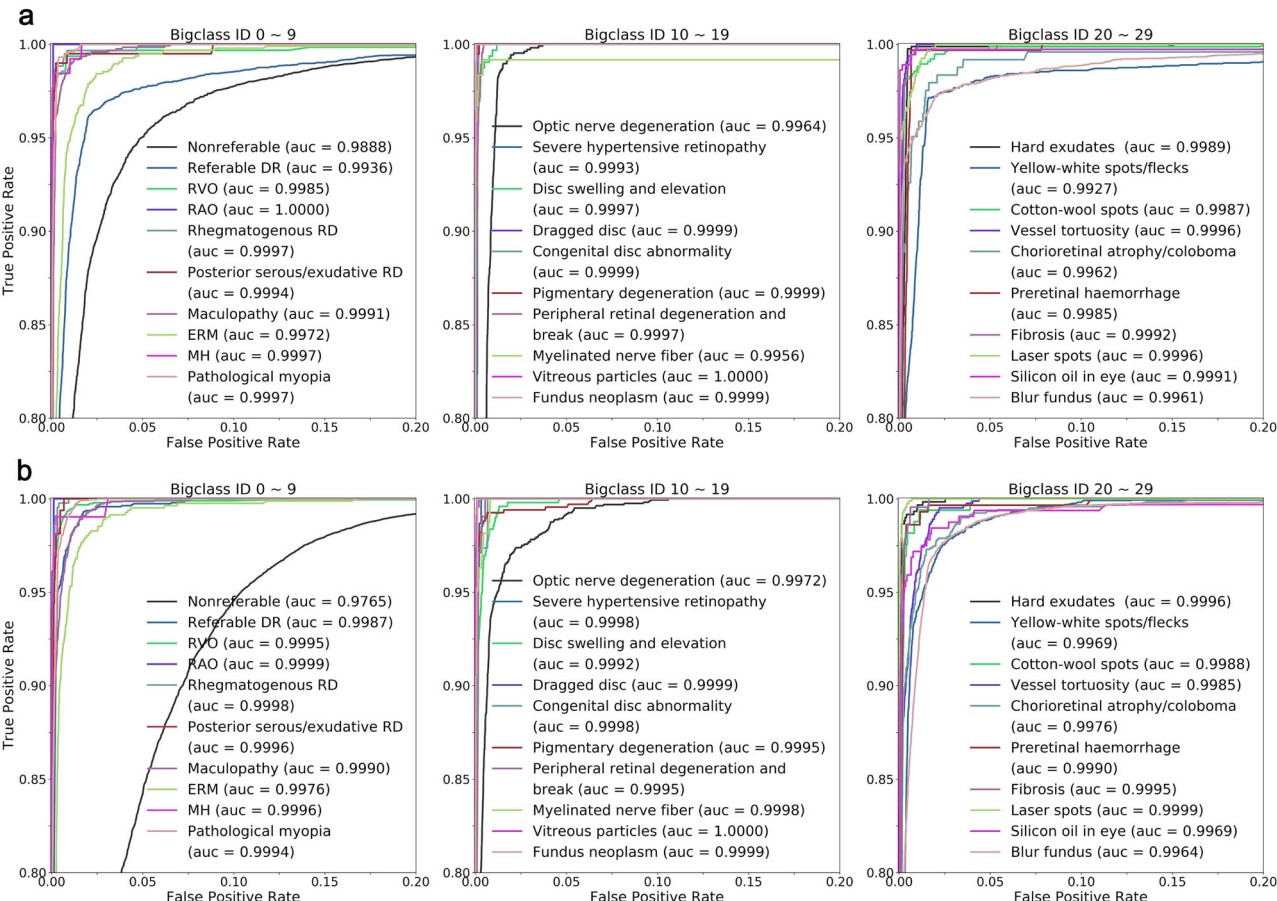

**Fig. 2 ROC and AUC of DLP for detection of bigclasses in primary test and multihospital test data sets. a** ROC curves and AUC for detecting every bigclass in primary test were calculated and plotted. Different colors of ROC curves corresponding to different AUC of each disease and condition are listed. **b** ROC curves and AUC for detecting every bigclasses in multihospital test. Source data are provided as a Source Data file.

frequency-weighted average $F_1$ score of 0.954, sensitivity of 0.943, specificity of 0.998, and subset accuracy of 92.45%. The DLP obtained a referable frequency-weighted average $F_1$ score of 0.964, sensitivity of 0.972, specificity of 0.998, and subset accuracy of 92.19%. Though subset accuracy is lower than the retina specialists, the DLP achieved higher average $F_1$ and sensitivity. The DLP was more sensitive than human experts on detecting multiple diseases. However, higher sensitivity for single-label images may lead to lower subset accuracy. For the JSIEC data set with patient information, the DLP obtained a referable frequency-weighted average $F_1$ score of 0.961, sensitivity of 0.968, specificity of 0.998, and subset accuracy of 91.28%. The average performance of retina specialists with patient information (referable frequency-weighted average $F_1$ score of 0.960, sensitivity of 0.95, specificity of 0.998, and subset accuracy of 92.91%) was enhanced than that without patient information. Higher subset accuracy than the DLP was obtained. The performance of the DLP was comparable to that of the retina specialists who had more than 10 years of clinical experience.

**Tele-reading applications.** To verify the automatic detection efficiency of our DLP for fundus diseases in real-life setting, tele-reading applications were conducted in 7 primary hospitals or community health centers located in different parts of China (Fig. 4). To avoid misjudgment of quality control during the prediction process, all images were classified into 30 bigclasses, regardless of the image quality score. Accordingly, images with low quality score were mainly triaged as blur fundus. Totally 7529

images from 5159 eyes and 3610 subjects were uploaded by the seven hospitals, among which 1362 images (538 eyes and 341 subjects) were merely detected as "Blur fundus" with probabilities equal to or larger than 95%. They were automatically sent back for repeat photography by the DLP. In addition, 1311 subjects (1832 eyes, 2105 images) were detected as nonreferable, which were also sent back directly to their primary hospitals with a suggestion of "follow-up". The rest (1958 subjects, 2789 eyes, 4062 images) were referable and further checked by the retina specialists, who confirmed 66 subjects (93 eyes, 106 images) to be nonreferable. After tele-reading application, these nonreferable cases were also reviewed by the retina specialists, and 11 subjects were then confirmed to be referable cases. These missed diagnosed cases were yellow-white spots (4 subjects), chorioretinal atrophy (1 subjects), and DR2 (6 subjects) with a few hemorrhage spots, which were not urgent for the referral. Thus, the DLP achieved a sensitivity of 0.994 and specificity of 0.952 for detection of referable in subject-based. However, there were 105 referable images that could not be categorized as any of the diseases and conditions listed in Supplementary Table 1 by the retinal specialists after the triage. They were omitted from the results of Supplementary Table 10. These images had rare conditions or unclear ophthalmic features. Details of classification results of categorized images for tele-reading application of our DLP in the seven hospitals for primary health care are summarized in Supplementary Table 10. The DLP achieved a referable frequency-weighted average F1 score of 0.913, sensitivity of 0.948, specificity of 0.997, and AUC of 0.9949. The subset accuracy for the tele-reading applications was 91.41%, comparable to

**Table 3 Performance of DLP in comparative test data set compared to experts (all fundus only, n = 922).**

| Diseases/conditions | ID | Average expert | | | | DLP | | | | |
|---|---|---|---|---|---|---|---|---|---|---|
| | | F1 | Sensitivity | Specificity | Subset accuracy[b],% | F1 | Sensitivity | Specificity | AUC (95% CI) | Subset accuracy, % |
| Non-referable | 0 | 0.875 | 0.941 | 0.990 | | 0.989 | 0.977 | 1.000 | 0.9955 (0.9877-1.0000) | |
| Referable DR | 1 | 0.951 | 0.935 | 0.993 | | 0.970 | 0.953 | 0.997 | 0.9942 (0.9894-0.9989) | |
| RVO | 2 | 0.954 | 0.951 | 0.996 | | 0.970 | 0.976 | 0.996 | 0.9984 (0.9964-1.0000) | |
| RAO | 3 | 0.942 | 0.929 | 0.999 | | 1.000 | 1.000 | 1.000 | 1.0000 (1.0000-1.0000) | |
| Rhegmatogenous RD | 4 | 0.980 | 0.989 | 0.999 | | 0.971 | 0.971 | 0.999 | 0.9996 (0.9991-1.0000) | |
| Posterior serous/exudative RD | 5 | 0.955 | 0.941 | 0.999 | | 0.964 | 1.000 | 0.998 | 0.9989 (0.9976-1.0000) | |
| Maculopathy | 6 | 0.951 | 0.955 | 0.996 | | 0.965 | 0.958 | 0.998 | 0.9967 (0.9929-1.0000) | |
| ERM | 7 | 0.957 | 0.946 | 0.998 | | 0.962 | 0.981 | 0.997 | 0.9974 (0.9940-1.0000) | |
| MH | 8 | 0.957 | 0.945 | 0.999 | | 0.939 | 0.939 | 0.998 | 0.9712 (0.9154-1.0000) | |
| Pathological myopia | 9 | 0.962 | 0.952 | 0.998 | | 0.992 | 0.984 | 1.000 | 0.9999 (0.9997-1.0000) | |
| Optic nerve degeneration | 10 | 0.961 | 0.950 | 0.998 | | 0.972 | 1.000 | 0.997 | 0.9996 (0.9988-1.0000) | |
| Severe hypertensive retinopathy | 11 | 0.895 | 0.885 | 0.997 | | 0.852 | 1.000 | 0.990 | 0.9961 (0.9919-1.0000) | |
| Disc swelling and elevation | 12 | 0.963 | 0.955 | 0.998 | | 0.970 | 1.000 | 0.997 | 0.9997 (0.9993-1.0000) | |
| Dragged disc | 13 | 0.932 | 0.882 | 1.000 | | 0.970 | 0.941 | 1.000 | 0.9849 (0.9553-1.0000) | |
| Congenital disc abnormality | 14 | 0.860 | 0.891 | 0.998 | | 0.952 | 0.909 | 1.000 | 0.9880 (0.9643-1.0000) | |
| Pigmentary degeneration | 15 | 0.989 | 0.978 | 1.000 | | 1.000 | 1.000 | 1.000 | 1.0000 (1.0000-1.0000) | |
| Peripheral retinal degeneration and break | 16 | 0.966 | 0.948 | 1.000 | | 1.000 | 1.000 | 1.000 | 1.0000 (1.0000-1.0000) | |
| Myelinated nerve fiber | 17 | 0.996 | 1.000 | 1.000 | | 1.000 | 1.000 | 1.000 | 1.0000 (1.0000-1.0000) | |
| Vitreous particles | 18 | 1.000 | 1.000 | 1.000 | | 0.966 | 1.000 | 0.999 | 1.0000 (1.0000-1.0000) | |
| Fundus neoplasm | 19 | 0.918 | 0.933 | 0.999 | | 1.000 | 1.000 | 1.000 | 1.0000 (1.0000-1.0000) | |
| Hard exudates | 20 | 0.959 | 0.994 | 0.997 | | 0.946 | 1.000 | 0.995 | 0.9998 (0.9994-1.0000) | |
| Yellow-white spots/flecks | 21 | 0.931 | 0.926 | 0.995 | | 0.898 | 0.971 | 0.985 | 0.9929 (0.9868-0.9989) | |
| Cotton-wool spots | 22 | 0.962 | 0.934 | 0.999 | | 0.982 | 0.966 | 1.000 | 0.9989 (0.9974-1.0000) | |
| Vessel tortuosity | 23 | 0.797 | 0.741 | 0.998 | | 0.867 | 0.765 | 1.000 | 0.9917 (0.9816-1.0000) | |
| Chorioretinal atrophy/coloboma | 24 | 0.951 | 0.922 | 0.999 | | 0.958 | 1.000 | 0.995 | 0.9980 (0.9958-1.0000) | |
| Preretinal hemorrhage | 25 | 0.949 | 0.958 | 0.998 | | 0.970 | 0.970 | 0.999 | 0.9932 (0.9799-1.0000) | |
| Fibrosis | 26 | 0.947 | 0.935 | 0.998 | | 0.976 | 1.000 | 0.998 | 0.9992 (0.9980-1.0000) | |
| Laser spots | 27 | 0.973 | 0.951 | 1.000 | | 0.964 | 0.930 | 1.000 | 0.9913 (0.9803-1.0000) | |
| Silicon oil in eye | 28 | 0.987 | 0.981 | 1.000 | | 1.000 | 1.000 | 1.000 | 1.0000 (1.0000-1.0000) | |
| Blur fundus | 29 | 0.833 | 0.750 | 1.000 | | 0.750 | 0.750 | 0.999 | 0.9499 (0.8525-1.0000) | |
| Referable[a], frequency-weighted average | | 0.954 | 0.943 | 0.998 | | 0.964 | 0.972 | 0.998 | 0.9964 | |
| Total | | | | | 92.45 | | | | | 92.19 |

[a]Referable refers to bigclass 1-29.
[b]Subset accuracy measures the scale of samples having identical labels between DLP prediction and the ground-truth labels.

that of the external multihospital test. These indicate the high efficiency of our DLP for triage of fundus diseases in the primary hospitals.

## Discussion

To our knowledge, this is the first report to show up to 39 types of eye diseases and conditions that can be detected by deep learning algorithms in retinal fundus photography at an accuracy level compatible with retina specialists. For single retinal disease, a number of studies conducted on DR screening reported high sensitivity (>90%), specificity (>90%), and AUC (>0.95)[27–29]. Ting et al.[18] reported a DLS developed and validated with about half a million images from multiethnic communities that were capable of screening not only DR but also possibly for glaucoma and AMD[18]. Automatic detection of ten classes of retinal fundus diseases has been reported, but with a small data set of images and low accuracy[30]. The capability of a DLP to detect multiple retinal diseases will greatly enhance the efficiency and cost-effectiveness of prompt diagnosis and treatment of patients especially in remote areas lacking ophthalmologists. This study provides a methodology with proven validation and tele-reading applications that would enhance ophthalmic service.

This DLP was trained with three diverse image data sets. Images of the JSIEC data set were obtained with consistently high quality of clear lesions, which ensured the DLP of learning the specific features of different diseases and conditions. The LEDRS data set[31] was collected from 13 hospitals across mainland China and stored images with varying qualities by different types of fundus camera. It, therefore, is useful for the analysis of diverse images. Images from EYEPACS[32], a publicly available data set collected mainly from Caucasians, together with other ethnic groups from different regions of the United States, further enhanced the diversities of the data sources.

In clinical screening application, a diagnostic procedure for single diseases would be able to detect only individual diseases. Using our DLP for multiple diseases and conditions, images triaged as nonreferable can be marked as cases with no need for further referral. On the other hand, images triaged as referable need additional confirmation by ophthalmologists. If the images are detected as disease classes, the ophthalmologists will make final diagnoses with references from the DLP. If the images are detected only as condition classes, the ophthalmologists will have to determine whether additional examinations are needed to confirm the diagnosis. Patients with unclear fundus can be detected as blur fundus for repeat photography or referral. Categorization based on common retinal diseases and fundus features enables detection of a wide spectrum of diseases, conditions, and unidentified diseases. If an image containing perceptible lesions cannot be classified as any of the pre-defined classes, it will be diagnosed as one of the referable classes. In addition, images with multiple morbidities can also be detected by our DLP in multi-label setting based on the assumption of independent probabilities of different classes.

The DLP that we have developed in this study is capable of automatically detecting almost all common fundus diseases and conditions (Supplementary Table 1) with a high $F_1$ score, sensitivity, and specificity. We obtained higher AUCs, mostly >0.996, in both primary tests and multihospital tests in multi-label setting (Table 2) when compared with those attained from the public data sets or with reported studies for detecting single diseases, with reported AUC ranging 0.889–0.983 for referable DR[18], 0.940 for large drusen[21], 0.986 for DR[33] and 0.986 for glaucomatous optic neuropathy[34]. Our apparently better performance was likely due to the extreme imbalance in the distribution of our data set. We used a multi-label setting, which included 30 bigclasses.

Labels were very sparse and most samples belonged only to one label. The class imbalance ratios ranged from 0.7 to 4028.7 in our data sets (Supplementary Fig. 5). Consequently, ROC and AUC can be over-optimistic and even unreliable in these situations[35,36]. High AUC results were also reported in a study using a multi-label setting with high imbalance ratios[37]. Therefore, $F_1$ scores should be more suitable for the evaluation of the performance of the algorithms for multi-label settings with extreme imbalance ratios. Another possible explanation is that during the labeling procedure, some uncategorized images were discarded because of poor image quality or uncertain features as determined by the retina expert panel[38]. The final results of AUC could thus be high.

To achieve a high $F_1$ score, sensitivity, and specificity, we applied a customized two-step strategy to construct deep learning algorithms as described in the "Methods" section. For bigclass classifications, features for different diseases and conditions, such as flame-shade hemorrhage, chorioretinal atrophy, retinal detachment, myelinated nerve fiber, and laser spots, were clear and obvious. Therefore, traditional models with medium resolution (299 × 299 pixels) should be adequate to obtain satisfactory results. Accordingly, this DLP achieved higher sensitivity and specificity than previous studies, which mainly focused on the grading of a single disease, such as DR[28] or AMD[39]. Large models with higher resolution (448 × 448 pixels) were used to separate DR1 from normal fundus images, which may differ in only one single microaneurysm. To distinguish possible glaucoma and optic atrophy, small models with a resolution of 112 × 112 pixels were applied after optic disc area segmentation and ROI cropping. In addition, we used dynamic data resampling and weighted cross-entropy loss function to resolve the imbalances of classes, while test time image augmentation and model ensembling have improved accuracy and robustness.

How a deep neural network makes predictions has been regarded as a black-box issue that may hinder clinicians to apply deep learning for clinical work[40]. In this study, we provide a modified CAM and DeepShap simultaneously to allow the interpretability of predicted results for every fundus image (Fig. 1). Though quantitative evaluation on the performance of heatmaps was difficult when there were multiple diseases and features, such facilities were capable to show how the DLP makes decisions by explicit fundus features including hemorrhages, exudates, hyperemia, and pale disc. Therefore, clinicians were able to "see" the lesion areas from the DLP and verify whether the DLP has used "appropriate" features for diagnosis. We have made quantitative analysis for bigclass 10 (optic nerve degeneration) true positive samples, which showed 100% (1,054 images) of their corresponding heatmaps were focused on the optic disc areas that were highly consistent with expert domain knowledge.

This study has several limitations. First, unlike common diseases such as DR, there is no universal diagnostic standard or consensus for reference of most fundus diseases if based only on fundus images. We, therefore, set up imaging reference standards according to fundus signs described in *EyeWiki* and textbooks, which were then reviewed by the retina expert panel for the final agreement. Moreover, there is also few publicly available standard image data sets of most fundus diseases for validation.

Second, though the DLP has been trained for up to 39 types of common diseases and conditions, it has not covered all fundus diseases and conditions. While common diseases, such as BRVO, CRVO, RD, DR, AMD, and glaucoma, with a large patient population and image data set, are convenient for deep feature learning, rare diseases are not practical to be given separate classes for deep feature pattern recognition. Though common feature-based classes partly render the detection of some rare diseases as nonreferable, other rare or unknown diseases (such as rare ocular tumors) remained difficult to be accurately identified

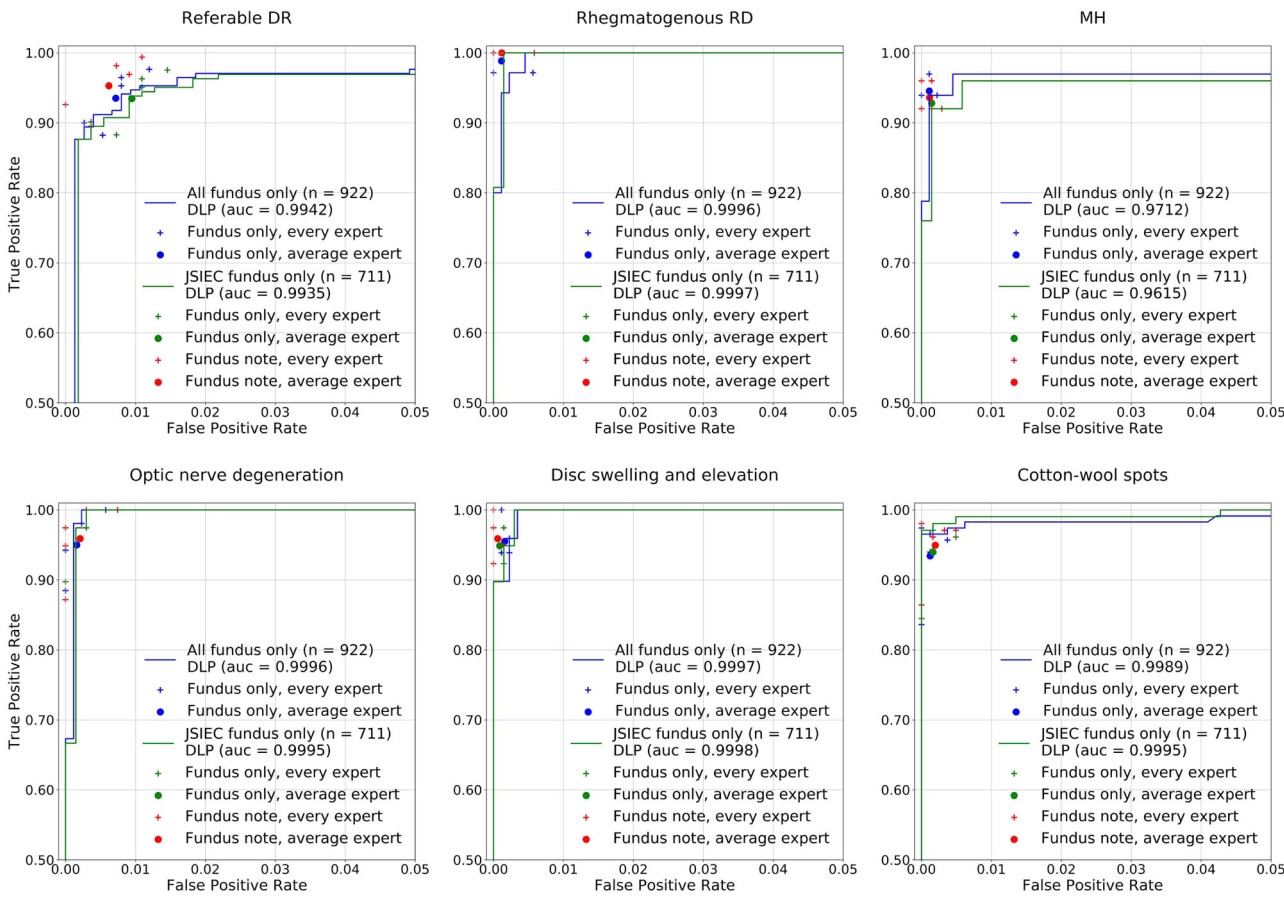

**Fig. 3 Performance of DLP in comparative test data set compare to retina experts.** Examples of ROC curves and AUC for detecting referable DR, rhegmatogenous RD, ERM, optic nerve degeneration, disc swelling and elevation, and cotton-wool spots in comparative test data set were calculated and plotted (blue curves for all images, and green curves for JSIEC images). Performance of individual retina expert (more than 10 years' clinical experiences in the retina specialty) is indicated by the crosses, and averaged expert performance by dots. Blue and green crosses/dots denote performance without patient information and red crosses/dots denote performance with patient information. Source data are provided as a Source Data file.

without sufficient image data for effective training. Adding an "others" class can be an effective way to combat this problem. However, due to high intra-class variation and low inter-class variations, it also reduces the ability to detect images of known classes.

The third limitation was the existence of overlaps between classes such as referable DR and Hard exudates. Multi-label classification based on Binary Relevance was used to tackle both label overlaps and class overlaps. In order to eliminate class overlaps, some of these 39 classes should be further decomposed.

Fourthly, only image classification and weakly image segmentation (used for CAM) were implemented in the DLP. Therefore, lesion areas such as exudates, hemorrhages, and cotton-wool spots could not be accurately located.

Fifthly, automatic diagnosis of DR1 is difficult since its only feature, microaneurysms, are presented as very small red dots even in high-resolution fundus images. Most previous studies did not report statistical results for DR1[18,27,28]. In a recent study on full grading of DR, results on DR1 detection were limited[41]. DR1 detection was unsatisfactory by our DLP, although we have customized designed big resolution classification models to cope with microaneurysms. Further work to detect DR1 by AI is warranted.

Lastly, this DLP provides diagnoses only based on fundus photography, which is just one component of comprehensive eye examinations. But it was designed and trained for the detection of multiple fundus diseases. Its performance would be less reliable than experts on diseases with subtle changes, such as referable DR only with tiny hard exudates or hemorrhage spot (Fig. 3). Confirmative diagnoses of fundus diseases, especially occult changes, require integrated medical history and full ophthalmoscopic examinations such as visual acuity, slit-lamp examination, OCT, and fundus fluorescence angiography.

In this study of retinal fundus images from multiple data sources with 39 types of fundus diseases and conditions, we have established a DLP with high $F_1$ score, sensitivity, specificity, and AUC for detection of multiple fundus diseases and conditions. It can be used in remote areas reliably and efficiently. We have thus extended the application of AI for one or several diseases to the whole spectrum of common fundus diseases and conditions, indicating the use of this platform for retinal fundus disease screening and triage, especially in remote areas around the world.

## Methods

**Classification**. All fundus images were labeled with 39 bigclasses/subclasses of diseases and conditions (Supplementary Table 1) according to the retinal signs described in *EyeWiki*, which is an online medical wiki encyclopedia launched in July 2010 by ophthalmologists with support by the American Academy of Ophthalmology[42], and cited in standard textbooks[43,44]. Common fundus diseases with distinct retinal characteristics recognizable by fundus images were classified as independent bigclasses. These included RVO, RAO, epiretinal membrane (ERM), rhegmatogenous retinal detachment (RD), macular hole (MH), pathological myopia, severe hypertensive retinopathy, peripheral retinal degeneration/break and myelinated nerve fiber. Some diseases shared similar characteristics and were not readily distinguishable according to fundus image only. For instance, hard exudate, sub-retinal hemorrhage, neovascularization, pigment epithelial detachment and

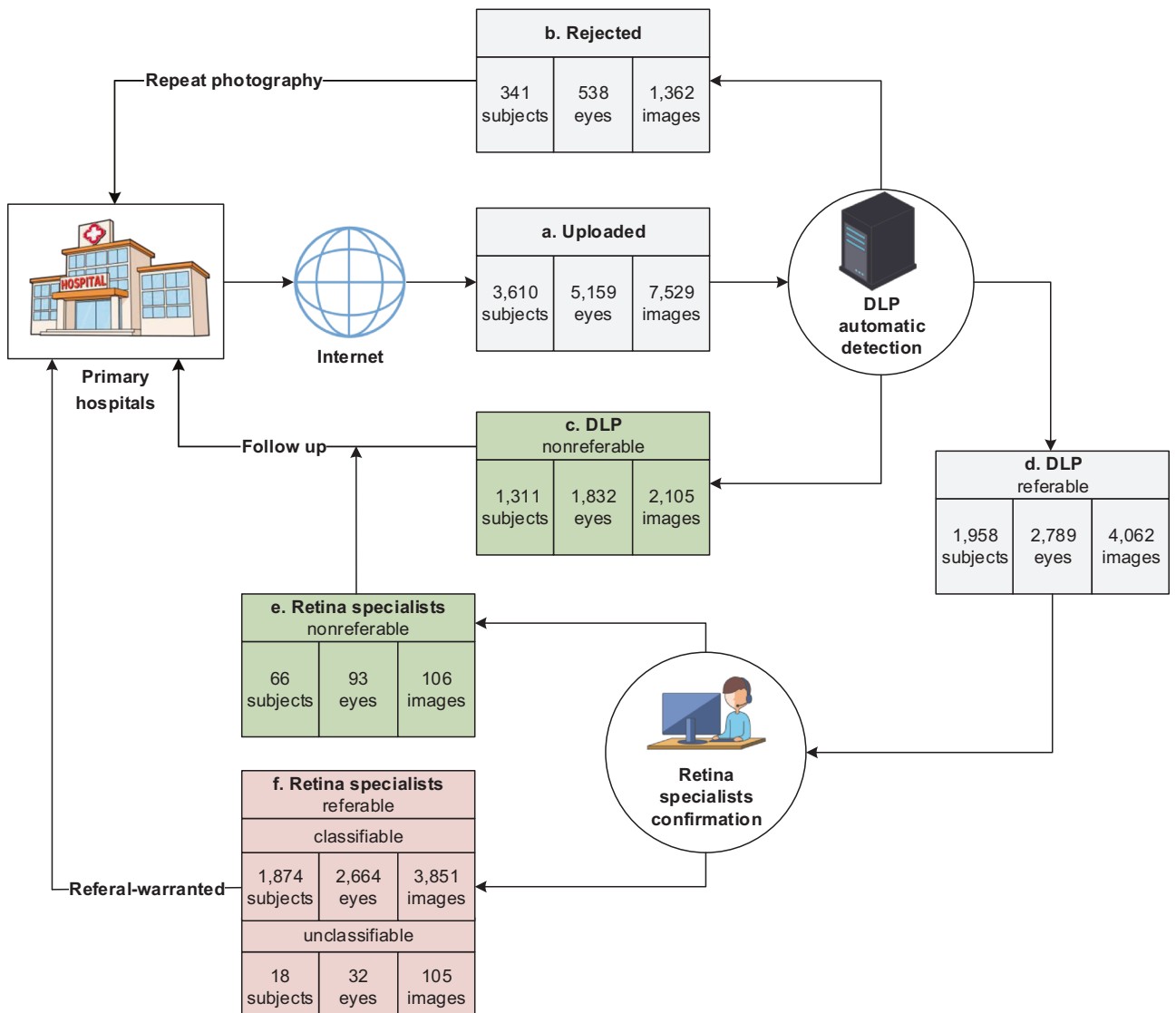

**Fig. 4 Tele-reading application of the DLP. a** Images were uploaded from seven hospitals for primary care located in different parts of China. **b** Images were rejected and sent back for repeat photography if merely detected as "Blur fundus" with probabilities equal to or larger than 95%. **c–f** Referral results were calculated in subject-based. Subjects were regarded as nonreferable if no image was detected as referable, otherwise, they were regarded as referable and sent for retina specialist confirmation. **c** Images were reviewed by retina specialists at the end of tele-reading application, among which 11 subjects were confirmed to be referable. **f** 1892 subjects with 3956 images were confirmed as referable, among which 3851 images were classifiable and 105 images could not be categorized as any of the diseases and conditions enlisted in Supplementary Table 1. Subjects were counted as unclassifiable only if all their images were judged as unclassifiable.

macular atrophy can be found in wet AMD, PCV, choroidal neovascularization, macular atrophy, retinal angiomatous proliferation, and idiopathic macular tel-angiectasia. We clustered them together and classified as the condition of macu-lopathy. Likewise, conditions causing optic disc swelling and elevation, such as papillitis, anterior ischemic optic neuropathy, papilledema, and pseudopapilloe-dema were classified as disc swelling and elevation. Other clustered conditions included optic nerve degeneration, posterior serous/exudative RD dragged disc, congenital abnormal disc, and fundus neoplasm. Some bigclasses were classified according to common features such as massive hard exudates, yellow-white spots/flecks, cotton-wool spots, vessel tortuosity, chorioretinal atrophy, coloboma, fibrosis, and preretinal hemorrhage. Cases of rare diseases with these features could be included in such bigclasses, which were thus sufficiently large in number for deep feature pattern recognition. Post-treatment conditions such as laser spots and silicon oil in the eye were included to recognize images of patients after surgery. It also helped to identify cases after surgery that required follow-up. Blurred fundus images, in which more than half of the image area could be obscured, were applied to train the deep learning algorithm, so that referrals for additional fundus pho-tography or transfer to ophthalmic specialists could be promptly arranged.

Bigclasses that might overlap with each other were restricted to specific criteria. For instance, ERM was restricted within the vessel arch, while the fibrosis must be

across or outside the vessel arch. DR grading was performed according to the guidelines for diabetic eye care by the International Council of Ophthalmology (ICO)[45,46]. All DR images were initially divided into non-referable DR (DR1, mild nonproliferative DR) and referable DR (RDR), which was defined as a severity level of moderate nonproliferative DR or worse and/or referable diabetic macular edema (DME). Moderate nonproliferative DR was further separated from RDR as DR2. Severe nonproliferative DR and proliferative DR were grouped as DR3. Referable DME was evaluated based on the presence of hard exudates at the posterior pole of the retinal images.

According to the referring criteria, patients with normal fundus, tessellated fundus, DR1, and large optic cup were not immediately referred to ophthalmologists. In addition, DR1 was not readily identified by the small models with lower resolution as the first step of diagnosis. Therefore, these conditions were clustered as a nonreferable class, and each of these conditions was defined as a subclass. In addition, diseases and conditions with similar signs were also clustered as a class, such as DR2 and DR3, BRVO and CRVO, CSCR and VKH disease, possible glaucoma and optic atrophy, retinitis pigmentosa (RP), and Bietti's crystalline dystrophy, as well as blurred fundus image with and without suspected PDR. After clustering, 39 types of diseases and conditions were divided into 30 bigclasses and 16 subclasses. A class identity number (ID) was given to each class

(Supplementary Table 1), which was composed of two parts: the number prior to the decimal point (0–29) denoting the 30 bigclasses, the number post to the decimal point denoting the subclasses.

**Data sets and labeling**. Fundus images from seven diverse data sources were collected for deep learning algorithm development and validation (Table 1). Image collection and the research protocol were in compliance with all relevant ethical regulations for studies involving human subjects and approved by the Human Ethics Committee of JSIEC, who also waived the informed consent from patients due to anonymity according to the Regulations for Ethical Review of Biomedical Research Involving Human of China. The primary data set for training, validation, and test was collected from the Picture Archiving and Communication Systems (PACS) at JSIEC in China, the Lifeline Express Diabetic Retinopathy Screening Systems (LEDRS) in China, and the Eye Picture Archive Communication System (EyePACS) Kaggle in the United States. The JSIEC data set was collected from PACS JSIEC between September 2009 and December 2018. The images were taken by a ZEISS FF450 Plus IR Fundus Camera (2009–2013) and Topcon TRC-50DX Mydriatic Retinal Camera (2013–2018) in a 35–50° field setting. The LEDRS data set had all posterior fundus images obtained from LEDRS, which is a multihospital-based program across mainland China[31]. The data were uploaded between April 2014 and December 2018 from 13 hospitals located in different parts of China (Guilin City Second Hospital, Jilin City Chinese Traditional and Western Medicine Hospital, Jinan City Lixia District People's Hospital, Luoyang City Third Hospital, Luzhou City Red Cross Hospital, Huhehaote Neimengu Province People Hospital, Zhanjiang City Second Hospital, Zhengzhou City Second Hospital, Beihai People's Hospital, Zhoukou City Eye Hospital, Nanyang City Ninth People's Hospital, Chongqing Wanzhou District People's Hospital, Liuzhou Red Cross Hospital). The EyePACS images were macular-centered fundus images obtained from the EyePACS public data set (EyePACS LLC, Berkeley, CA)[32], which is a tele-reading program for diabetic retinopathy screening in community clinics across the United States. These images were acquired from different makes and models of cameras with varied features and specifications. Such variations in input images enhance the generalization capability of the algorithm.

Primary data set acquisition and processing flow are shown in Supplementary Fig. 1 and Supplementary Fig. 2. Fundus images from JSIEC were exported and primarily classified by searching PACS with the nomenclature of fundus diseases listed in Supplementary Table 1. Images collected from LEDRS[31] were exported according to their graded records (five grades) marked by trained retina specialists through an internet-based system, followed by clustering into four classes based on the DR grading criteria as described previously. All images collected from various data sources were initially screened by an automatic quality control algorithm. Images scoring lower than 80 (0–100) were discarded. Totally 41,056 images (JSIEC 9126, LEDRS 16,523, EYEPACS 15,407) were excluded before the labeling procedure. To ensure the accuracy of classification by the deep learning algorithms, the images were labeled manually by 20 licensed ophthalmologists in China before algorithm training, validation, and testing. The ophthalmologists were separated into ten groups. Each group had a senior retina specialist with more than 7 years' clinical experience and an unspecialized ophthalmologist having trained for over 3 years. Images were initially labeled by unspecialized ophthalmologists, then confirmed by senior retina specialists. Images that were agreed by the specialists were applied for deep learning directly. Otherwise, images were transferred to a retina expert panel of five senior retina specialists for final decision. Besides, we have added referable labels (observation, routine, semi-urgent, urgent) to all categories of diseases and conditions. For "blur fundus", an additional suggestion for "repeat photography" was given (Supplementary Table 1). Unclassifiable images were excluded, which were firstly judged by unspecialized ophthalmologists and senior retina specialists and further confirmed by the retina experts (Supplementary Table 3). They either had poor image quality or uncertain features[38], or were rare diseases that did not belong to the 39 categories (Supplementary Fig. 2).

After labeling, image data from JSIEC and LEDRS were divided into two parts according to the collecting date. The part before 2018 was randomly split into a training set (85%) and validation set (15%) of specific patients. Therefore, no image from the same patient appeared in both the training and the validation data sets. The other part within 2018 was applied as a test data set, which has excluded those cases imaged before 2018. Same partition in patient-based for images from EyePACS was also conducted but randomly without information of the collection date. Apart from the primary test, the DLP was further tested with three external hetero-ethnic data sets from different parts of China: Fujian in southeastern China (Xiamen Kehong Eye Hospital), Tibet in western China (Mentseekhang, Traditional Tibetan Hospital), and Xinjiang in northwestern China (DuShanZi People's Hospital). In Fujian, all study subjects were of Han ethnicity, while in the Tibet and Xinjiang data sets there were 97.1% Tibetan and 13.3% Uyghur ethnicity, respectively. The labeling procedure was the same as that for the primary data sets. Furthermore, the DLP had been tested with four public data sets in a single-disease setting: Messidor-2, Indian Diabetic Retinopathy Image Data set (IDRID), PALM, and Retinal Fundus Glaucoma challenge (REFUGE)[26]. The Messidor-2 data set was collected from four French

eye institutions, which has been widely used for benchmarking the performance of automatic diabetic retinopathy detection[27,47]. The majority rule was applied for labeling the Messidor-2 data set by the retina expert panel. The IDRID data set was established for "Diabetic Retinopathy: Segmentation and Grading Challenge" workshop at IEEE International Symposium on Biomedical Imaging (ISBI-2018). PALM was a challenging event of ISBI-2019 that focused on the development of algorithms for the diagnosis of pathological myopia by fundus photos. REFUGE was a competition as part of the Ophthalmic Medical Image Analysis (OMIA) workshop at MICCAI 2018, which provided a data set of fundus images with clinical glaucoma labels based on a comprehensive evaluation of clinical records, including follow-up fundus images, IOP measurements, OCT images, and visual fields[26].

To evaluate the efficiency of our DLP in actual application for fundus diseases triage, tele-reading applications were conducted in seven primary hospitals for primary health care or community health centers between July 2019 and May 2020. These hospitals or centers belonged to the JSIEC-Specialized Treatment Combination (STC), which included Balinzuoqi Hospital of Traditional Mongolian Medicine and Chinese Medicine (Inner Mongolia), Hainan Tibetan Autonomous Prefecture People's Hospital (Qinghai), Nyingchi People's Hospital (Tibet), DuShanzi People's Hospital (Xinjiang), Sanrao Community Health center (Guangdong), Huizhai Community Health center (Guangdong), Zhongshan Torch Development Zone Hospital (Guangdong). Images from these hospitals for primary health care were sent through the Internet to JSIEC, which were then automatically detected by the DLP and a graphic report was generated. Reports were directly sent back to the primary hospitals if no referable disease was detected. Suggestions of "rephotograph" were also given if images were merely detected as "Blur fundus" with probabilities equal to or larger than 95%. Otherwise, images and the corresponding reports were sent to retina specialists for confirmation before sending back to the JSTC hospitals. Processing flow for tele-reading application are shown in Fig. 4. To obtain the statistical results, all upload images were reviewed by the retina specialists after the triage in the tele-reading applications.

In this study, totally 249,620 images were included for training, validation, and tests, image data sets distributions in various bigclasses and subclasses are shown in Supplementary Tables 5–11. The data sets show extreme imbalance and the labels were very sparse in some bigclasses, the imbalance ratio of negative samples vs positive samples ranged from 1.8 to 956.5 in the primary training data sets (Supplementary Fig. 5)[48].

**Architecture of the DLP**. The whole data flow and simplified architecture of the DLP are shown in Supplementary Figs. 1 and 4. Accompanied with the image processing methods, 3 groups of CNNs and a Mask-RCNN (Supplementary Table 4) were applied to construct a two-level hierarchical system for the classification of the 39 types of diseases and conditions. Image quality of each input image was initially evaluated. After being further preprocessed, fundus images were then applied to the multi-label classification of 30 bigclasses. Images classified to bigclass 0, 1, 2, 5, 10, 15, and 29 were further classified to subclasses with CNNs. To increase the diagnostic accuracy, the optic disc areas of images for bigclass 10 were identified and cropped out for multi-class classification (2 classes) of possible glaucoma and optic atrophy using customized designed CNNs (Supplementary Fig. 3) for small input image size (112 × 112). During the training processes, dynamic data resampling (Algorithm 1, Supplementary information), real time image augmentation and transfer learning were used. Probability values and two types of heatmaps, Class activation maps (CAM) and Deepshap, were obtained by CNNs. After being fully trained and validated, all CNNs models were deployed into production environment for internal testing. Technical details of algorithms and implementation are explicated in the Supplementary Methods: "Algorithm development and the DLP deployment" section.

**Statistical analysis**. Statistical analysis was performed on bigclasses and subclasses separately to evaluate the performance of the DLP. For every class in the primary data sets (Training, Validation, and Test), multi-label classification evaluation was conducted by obtaining the true negative (TN), false positive (FP), false negative (FN), true positive (TP), $F_1$ score, sensitivity, specificity at binary decision thresholds, and calculating the ROC analysis and AUC by giving their 95% confidence intervals (CIs)[13,48]. For the aggregate measure of the DLP performance on referable bigclasses, we computed the class frequency-weighted arithmetic mean for the $F_1$ score, sensitivity, specificity, and the AUC[13]. Two-sided 95% CIs were calculated with Delong's method for AUC using the open-source package pROC (version 1.14.0)[49]. We evaluated the detection accuracy on every image by calculating the subset accuracy, which provided the scale of samples having identical labels between DLP prediction and the ground-truth labels[48]. Binary classification analysis of subclasses (ID 0, 1, 2, 5, 10, 15, 29) were assessed to obtain similar statistical parameters to multi-label classification[13,48]. Three parallel binary classifications in nonreferable (four subclasses) were conducted to detect the probabilities of the tessellated fundus, large optic cup, and DR1, respectively, against the normal subclass. Performance of the multi-label classification was further tested with external multihospital data sets (Fujian, Tibet, and Xinjiang), the publically available data sets (Messidor-2, IDRID, PALM, and REFUGE), retina expert comparison, and tele-reading applications.

**Reporting summary**. Further information on research design is available in the Nature Research Reporting Summary linked to this article.

## Data availability

Data set from EYEPACS Kaggle for training, validation, and test is available at https://www.kaggle.com/c/diabetic-retinopathy-detection. Public data sets for tests are available at Messidor-2: https://www.adcis.net/en/third-party/messidor2/, IDRID: https://idrid.grand-challenge.org/, REFUGE: https://refuge.grand-challenge.org/, PALM: https://palm.grand-challenge.org/. Other data sets supporting the findings of the current study are not publicly available due to the confidentiality policy of the National Health Commission of China and institutional patient privacy regulation. But they are available from the corresponding author upon request. The data can be accessed for the purpose of reproducing the results and/or further academic and AI-related research activities from the corresponding author M.Z. upon request within 10 working days. Source data are provided with this paper.

## Code availability

The source codes are available on Github: for image preprocessing, https://github.com/linchundan88/Fundus-image-preprocessing[50]; for web application (https://github.com/linchundan88/fundus_multiple_diseases_web[50]); and for training, validation, test and RPC (remote procedure call) service, https://github.com/linchundan88/fundus_multiple_diseases[50].

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

## Acknowledgements

This study was supported by the National Natural Science Foundation of China (NSFC, 81570849 to L.-P.C., 81800822 to S.T.), Natural Science Foundation of Guangdong Province, China (NSFG, 2020A1515010415 to L.-P.C.), Grant for Key Disciplinary Project of Clinical Medicine under the Guangdong High-level University Development Program (002-18119101), LKSF cross-disciplinary research grants (2020LKSFG16B to L.-P.C. and M.Z.), and an internal grant from the Joint Shantou International Eye Center of The Shantou University and The Chinese University of Hong Kong (17-003 to L.-P.C.).

## Author contributions

L.-P.C. and J.-W.L. initiated the project and the collaboration. J.Ji. developed the network architectures, training, and testing setup. L.-P.C. and M.Z. designed the clinical setup. L.-P.C. and S.-T.J. defined the clinical labels. J.-W.L. and J.Ji. contributed to software engineering. L.-P.C., S.-T.J., H.-J.L., T.-P.L., Yun.W., J.-F.Y., Y.-F.L., L.T., D.L., Y.W., D.Z., Y.X., H.W., J.J, Z.W., D.H., T.S., B.C., J.Y., X.Z., L.L., C.H., S.T. and Y.H. labeled the databases. L.-P.C., G.Z., H.C., W.C., and M.Z. contributed to clinical assessments. L.-P.C. and J.-W.L. analyzed the data. L.-P.C. and C.-P.P. interpreted the data. L.-P.C. and M.Z. managed the project and obtained funding. L.-P.C., J.Ji., J.-W.L., and S.-T.J. wrote the paper. L.-P.C., J.-W.L., J.Ji., T.-K.N., and C.-P.P. critically revised the manuscript for important intellectual contents. L.-P.C. and J.Ji. contributed equally.

## Competing interests

The authors declare no competing interests.
