## [Peer Review File · Nature Communications]

REVIEWER COMMENTS

Reviewer #1 (Remarks to the Author):

Large scale prospective studies conducted in real world settings (like the Tele-reading application in this study) are important to make the case for adoption of AI in healthcare. I commend the authors for undertaking one. That said, I do have some concerns about publishing this manuscript as is written.

My main concern about this manuscript is that I do not see any methodological novelty in the development of the deep learning system, and yet the reported AUCs are much higher than reported by others. Do the authors have a good explanation for why that is the case? The average AUC in Table 2 is 0.999 with very tight confidence intervals in most cases. As a comparison [20] reports an AUC in the 0.889-0.983 range for referable DR [23] reports an AUC of 0.94 for detecting large drusen Li. et al, Ophthalmology 2018, report an AUC of 0.986 for Glaucomatous Optic Neuropathy Krause et al, Ophthalmology 2018, report an AUC of 0.986 for referable DR Some of the above examples claim model performance exceeding that of experts, whereas in this study (Figure 3) the model performance is worse than experts at least on some of the predictions (most notably in referable DR) Can the authors validate their DLP on publicly available datasets for at least some of the predictions and report the performance for comparison? Two example public datasets for validating the referable DR predictions are EyePACS Kaggle dataset: <https://www.kaggle.com/c/diabetic-retinopathy-detection/overview> Messidor-2 Kaggle dataset: <https://www.kaggle.com/google-brain/messidor2-dr-grades>

Just to clarify, I do think this study has merit despite no novelty in the deep learning methods applied, as the main objective is not advancing deep learning methods. My skepticism is mainly around the claimed performance of the deep learning platform.

Other questions

Line 226. How can this happen (i.e. 105 referable images not categorized) as the algorithm described in supplement Line 124 will select one of the classes based on the maximum probability?

Line 450 Is the "quality control algorithm" referred to here the same as what is described under "Fundus image quality assessment" in the supplement? Please include some statistics on how many images got discarded by the quality control algorithm. How is the quality control algorithm different from the "blur fundus" classification?

There is no mention of either the "quality control algorithm" or "image quality assessment" in either the "Tele-reading applications" section or Figure 4. How is image quality assessed in that setting? Or do you only use the "blur fundus" classification for the tele-reading application?

Line 458 Statistics on intergrader agreement between the unspecialized ophthalmologists, between the unspecialized ophthalmologists and group specialists, between the unspecialized ophthalmologists and expert panel, and between the group specialists and expert panel should be included for each dataset, in order to understand how difficult the task was for human experts.

Line 466 What happens if a patient was imaged both before 2018 and after 2018. Do you discard such patients from the test set?

Line 471 How were the multihospital test, public test, and comparative test datasets labeled? Using the same process as described in Lines 452-462?

Figure 1 caption in main text should be for supplementary Figure 1, Figure 2 caption should be for Figure 1, Figure 3 caption for Figure 2, etc

Figure 4. Doesn't show triaging of nonreferable cases (as described in Line 220)

Reviewer #2 (Remarks to the Author):

The authors are to be commended for undertaking the monumental task for curating a dataset of over 250,00 images. However, there are a number of major and minor issues with this work. The manuscript was also quite challenging to read as a lot of the key information was in the supplementary section.

Major

- The 39 diseases are not really diseases but a mix of findings and diseases.
- Laser spots, silicon[e] oil etc are more a consequence of treatment!
- The external test sets were all DR, thus claiming that the performance on external datasets is perhaps overoptimistic
- There is expected to be a lot of overlap between the categories. It is not clear how this overlap was dealt with.
- Given that the architecture is not particularly novel, the performance of the algorithm seems extraordinarily high, given the current state of the art. Can you comment on what you believe has led to such high performance?
- The github code suggest the use of an optic disk localizer based on the REFUGE challenge -was this used in this work?
- Inclusion/exclusion of images without consensus labels should be clarified
- Can you confirm that all ~250,000 images were manually rated by 20 ophthalmologists? Or clarify that process?
- The utility and accuracy of the heatmaps are unclear. There is not quantification of their ability to localize.

Minor

- "bigclasses" is a unusual term
- not sure what "The sum of false negative (FN) and true positive (TP) represents the samples of corresponding class." means -can you clarify?

Reviewer #3 (Remarks to the Author):

This is a major study reporting on the development and validation of an AI system to autonomously detect a range of retinal conditions in fundus photographs. The authors amassed a very extensive database of retinal images that was labelled manually for 39 classes. Testing and validation was carried out in separate data sets, as well as external validation in independent datasets from the same region as well as public data sets. Furthermore, the system was tested in a clinical tele-reading setting, where it showed comparable performance to retina specialists.

The paper is the result of a huge interdisciplinary effort and at times challenging to navigate. For instance, when reading the paper, it is not entirely clear what "bigclass" is referring to, although this is important for understanding.

From a clinical standpoint I may offer a few major comments:

- 1) The 39 classes are a heterogeneous group of diagnoses (e.g. DR), signs (e.g. preretinal hemorrhage, disc swelling) and even postoperative findings (e.g. silicone oil in the eye). This appears as a collection of findings that were present in a certain dataset, and not as list of conditions that one would screen for in a medically oriented screening setting.
- 2) When compared to a benchmark paper in the field that had a similar goal, although using a different image modality (OCT, reference # 33 De Fauw et al Nature Medicine), what I am missing most in this current paper is a triage of referable conditions (urgent, routine, etc.). For instance, retinal detachment is much more urgent than myelinated nerve fiber layer (which in fact is a congenital condition without clinical relevance, but listed as referable here). The presented work only provides non-referable vs. referable classes.
- 3) The ground truth labels were provided by 20 ophthalmologists in 10 teams of each a retina specialist and an ophthalmologist in training. Did the authors study how reproducible these 10 teams worked compared to each other and along their task (inter-grader and intra-grader reproducibility)? I did not see these data presented.

REVIEWER COMMENTS

Reviewer #1 (Remarks to the Author):

Large scale prospective studies conducted in real world settings (like the Tele-reading application in this study) are important to make the case for adoption of AI in healthcare. I commend the authors for undertaking one. That said, I do have some concerns about publishing this manuscript as is written.

Response: Thank you very much for the comments which we have followed in revising the manuscript.

My main concern about this manuscript is that I do not see any methodological novelty in the development of the deep learning system, and yet the reported AUCs are much higher than reported by others. Do the authors have a good explanation for why that is the case? The average AUC in Table 2 is 0.999 with very tight confidence intervals in most cases. As a comparison

[20] reports an AUC in the 0.889-0.983 range for referable DR

[23] reports an AUC of 0.94 for detecting large drusen

Li. et al, Ophthalmology 2018, report an AUC of 0.986 for Glaucomatous Optic Neuropathy

Krause et al, Ophthalmology 2018, report an AUC of 0.986 for referable DR

Response: We agree that we worked on a conventional rather than novel approach, though we have used dynamic data-resampling and weighted cross entropy loss function simultaneously to cope with the bias prediction in class imbalance situations. We also used both CAM and DeepShap heatmaps to provide a certain degree of explainability. Our methodology was not very different from reported studies but we have attained better performance in AUC. But we think AUC can be over optimistic and unreliable in situations with huge imbalance ratios. F_1 should be more suitable for evaluating the performance of the DLP, but we kept providing AUC results for comparison. There should be two main reasons for our higher AUCs than those reported by others, different distribution of datasets and discarding of unclassifiable images and images with uncertain features in the labelling procedure. We have given explanation in the revised text with additional references.

Page 14, lines 278-293: "We obtained higher AUCs, mostly greater than 0.996, in both primary tests and multihospital tests in multi-label setting (Table 2) when compared with those attained from the public datasets or with reported studies for detecting single diseases, with reported AUC ranging 0.889-0.983 for referable DR¹⁹, 0.940 for large drusen²², 0.986 for DR³⁸ (Krause J. et al, Ophthalmology 2018). and 0.986 for glaucomatous optic neuropathy³⁹ (Li Z. et al, Ophthalmology 2018). Our apparently better performance was likely due to the extreme imbalance in the distribution of our dataset. We used multi-label setting, which included 30 bigclasses. Labels were very sparse and most samples belonged only to one label. The class imbalance ratios ranged from 0.7 to 4028.7 in our datasets. Consequently, ROC and AUC can be over optimistic and even unreliable in these situations^{40,41} (#40. Japkowicz N. and Holte R. 2000; #41. Saito T. and Rehmsmeier M. 2015). High AUC results were also reported in a study using multi-label setting with high imbalance ratios⁴² (Zhu H. et al. 2020). Therefore, F_1 scores should be more suitable for evaluation of performance of the algorithms for multi-label setting with extreme imbalance ratios. Another possible explanation is that during the labeling procedure, some uncategorized images were discarded because of poor image quality or uncertain features as determined by the retina expert panel⁴³(Son J. et al.2020). The final results of AUC could thus be high."

Some of the above examples claim model performance exceeding that of experts, whereas in this study(Figure3) the model performance is worse than experts at least on some of the predictions (most notably in referable DR)

Response: Our DLP is designed and trained for detection of multiple fundus diseases. Without specific optimization to certain single diseases, it is possible that our performance might be weaker than experts on some diseases with subtle changes. We admitted this as a limitation in the revised Discussion.

Page 17, lines 353-357: “Lastly, this DLP provides diagnoses only based on fundus photography, which is just one component of comprehensive eye examinations. But it was designed and trained for detection of multiple fundus diseases. Its performance would be less reliable than experts on diseases with subtle changes, such as referable DR only with tiny hard exudates or hemorrhage spot (Figure 3).

Can the authors validate their DLP on publicly available datasets for at least some of the predictions and report the performance for comparison? Two example public datasets for validating the referable DR predictions are

EyePACS Kaggle dataset: <https://www.kaggle.com/c/diabetic-retinopathy-detection/overview>

Messidor-2 Kaggle dataset: <https://www.kaggle.com/google-brain/messidor2-dr-grades>

Response: Many thanks for the very important comments. We have verified our DLP with publicly available datasets for comparison including Messidor-2, REFUGE, PALM and IDRiD datasets (EyePACS Kaggle dataset had been used for developing our DLP). Results has been added in the revised manuscript.

Page 5, lines 87-90: “4) Four publicly available datasets (n= 3,438): Messidor-2 (n=1,748), Indian Diabetic Retinopathy Image Dataset (IDRID) (n=516), Pathological Myopia (PALM) (n= 374), and Retinal Fundus Glaucoma challenge (REFUGE) (n=800).”

Page 8, lines 152-174: “After testing with hetero-ethnic datasets in China, the generalization capabilities of our DLP to detect different diseases and conditions were evaluated with 4 public test datasets in the single disease setting, messidor-2, IDRiD, PALM, and REFUGE (Supplementary Table 8). For detecting referable DR, we achieved a F1 score of 0.944, sensitivity of 0.906, specificity of 0.996 and AUC of 0.9861 in messidor-2. Performance was weaker in IDRiD with F1 score of 0.875, sensitivity of 0.824, specificity of 0.902 and AUC of 0.9431. We reviewed the misjudged cases and found presence of stains on dirty lens looking like hemorrhage spots as the main cause of false positive results. For pathological myopia, higher performance in PALM was achieved with a F1 score of 0.974, sensitivity of 0.958, specificity of 0.988 and AUC of 0.9931. Performance was moderate for optic nerve degeneration (possible glaucoma) with a F1 score of 0.651 (0.674), sensitivity of 0.850 (0.813), specificity of 0.915 (0.933) and AUC of 0.9397 when compared to the top 12 contestant teams in the REFUGE Challenge 28. Labels of all images in the FEFUGE dataset were initially confirmed by multiple examinations including intraocular pressure (IOP), optical coherence tomography (OCT) and visual field. In early-stage glaucoma, almost no noticeable change could be detected by fundus though OCT could show retinal nerve fiber layer thinning. These cases were missed by the DLP which was developed based on fundus images only. Without specific optimization to the datasets of single diseases for competitions, the overall performance of our multi-label DLP with single disease datasets in different distribution was acceptable. These indicate good generalization capabilities in our DLP for detecting fundus diseases in heterogeneous images.”

Just to clarify, I do think this study has merit despite no novelty in the deep learning methods

applied, as the main objective is not advancing deep learning methods. My skepticism is mainly around the claimed performance of the deep learning platform.

Response: Thank you very much for the encouraging and helpful comments.

Other questions

Line 226. How can this happen (i.e 105 referable images not categorized) as the algorithm described in supplement Line 124 will select one of the classes based on the maximum probability?

Response: You are right, the algorithm is based on the CWA (Close-World assumption). If an image is predicted to be negative for all classes, it will be given the class with maximum probability instead of predicted as unknown. All uploaded images were firstly triaged into any of the listed bigclasses, then followed with the checking by the retina specialists. Lastly, 105 images were confirmed as uncategorized ones by the retina specialists and omitted from the results of supplementary Table 8. We have attempted to clarify in the Results.

Page 12, lines 229-233: "However, there were 105 referable images that could not be categorized as any of the diseases and conditions listed in Supplementary Table 1 by the retinal specialists after the triage. They were omitted from the results of supplementary Table 8. These images had rare conditions or unclear ophthalmic features".

Page 25, lines 520-522: "To obtain the statistical results, all upload images were reviewed by the retina specialists after the triage in the tele-reading applications."

"

Line 450 Is the "quality control algorithm" referred to here the same as what is described under "Fundus image quality assessment" in the supplement? Please include some statistics on how many images got discarded by the quality control algorithm. How is the quality control algorithm different from the "blur fundus" classification?

There is no mention of either the "quality control algorithm" or "image quality assessment" in either the "Tele-reading applications" section or Figure 4. How is image quality assessed in that setting? Or do you only use the "blur fundus" classification for the tele-reading application?

Response: Sorry for the confusion. "quality control algorithm" in line 450 is the same as "Fundus image quality assessment" in the supplement. We have given clarification in the revised Methods.

Page 22, lines 458-461: "All images collected from various data sources were initially screened by an automatic quality control algorithm. Images scoring lower than 80 (0-100) were discarded. Totally 41,056 images (OH 9,126, LEDRS 16,523, EYEPACS 15,407) were excluded before the labeling procedure."

Quality control algorithm has been applied to exclude images of low quality due to inappropriate shooting, overexposure, underexposure or out of focus. Blur fundus mainly referred to images showing vague fundus usually caused by cataract, vitreous opacity or bleeding. In some cases, it may be difficult to distinguish between them.

Page 11, lines 212-214: "To avoid misjudgment of quality control during the prediction process, all images were classified into 30 bigclasses, regardless of the image quality score. Accordingly, images with low quality score were mainly triaged as blur fundus." Therefore, image quality assessment was not show in the Tele-reading application flow.

Line 458 Statistics on intergrader agreement between the unspecialized ophthalmologists, between the unspecialized ophthalmologists and group specialists, between the unspecialized ophthalmologists and expert panel, and between the group specialists and expert panel should be included for each dataset, in order to understand how difficult the task was for human experts.

Response: Thanks for the important comment. We have included the statistics of inter-grader comparison in the supplementary document. We have added "Supplementary Table 2", and mentioned in the Results.

Page 5, lines 77-78: "Inter-grader agreements in each datasets were analyzed (Supplementary Table 2)."

Line 466 What happens if a patient was imaged both before 2018 and after 2018. Do you discard such patients from the test set?

Response: Sorry for being unclear. We have discarded the images (2,677 images, 1,635 subjects) from the dataset of 2018 if the patients had been imaged before 2018. Clarification in the revised Methods.

Page 23, lines 479-481: "The other part within 2018 was applied as test dataset, which has excluded those cases imaged before 2018".

Results for the primary test set have been recalculated. We had missed calculated 1724 subjects in the primary OH dataset at the beginning. The corrected results are in Table 1.

Line 471 How were the multihospital test, public test, and comparative test datasets labeled? Using the same process as described in Lines 452-462?

Response: The multihospital test datasets were labeled using the same process as the primary datasets. Other public datasets (IDRID, REFUGE and PALM) were applied with their original labels for competitions. It is clarified in the Methods in the revised manuscript:

Page 24, lines 488-491: "The labeling procedure was the same as that for the primary datasets. Furthermore, the DLP had been tested with four public datasets in single disease setting: Messidor-2, Indian Diabetic Retinopathy Image Dataset (IDRID), PALM and Retinal Fundus Glaucoma challenge (REFUGE)²⁸."

Page 24, lines 494-495: "The majority rule was applied for labeling the Messidor-2 dataset by the retina expert panel."

The Messidor-2 and comparative test datasets were labelled by the expert panel. The majority decision served as the reference standard for classification, i.e. an image was labeled as referable DR if 3 or more panelists graded it referable DR. We have revised the Results.

Page 10, lines 182-185: "These images had not been "seen" by the DLP and involved various challenging diseases and conditions, which were arranged by the retina expert panel. The majority decision served as the reference standard for classification."

Figure1 caption in main text should be for supplementary Figure1, Figure2 caption should be for Figure1, Figure3 caption for Figure2, etc

Response: We are sorry for the mistakes, which have been corrected accordingly.

Figure 4. Doesn't show triaging of nonreferable cases (as described in Line 220)

Response: In the tele-reading application, nonreferable cases (1,311 subjects shown in "c.") triaged by the DLP were returned to the primary hospitals and were also reviewed by retina specialists after the tele-reading application. The flow of tele-reading was illustrated in Figure 4 but the retina specialist review after the tele-reading procedure was not shown. Clarification has been given in the Results.

Page 11, lines 222-225: "The nonreferable cases (1,311 subjects) were triaged directly by the DLP and returned to their primary hospitals. After the tele-reading application, these nonreferable cases were also reviewed by retina specialists and 11 subjects were then confirmed to be referable cases."

Reviewer #2 (Remarks to the Author):

The authors are to be commended for undertaking the monumental task for curating a dataset of over 250,00 images. However, there are a number of major and minor issues with this work. The manuscript was also quite challenging to read as a lot of the key information was in the supplementary section.

*Major -- The 39 diseases are not really diseases but a mix of findings and diseases.
-- Laser spots, silicon[e] oil etc are more a consequence of treatment!*

Response: Many thanks for the correction. They are indeed a mix of diseases, findings, and conditions. Laser spots and silicon oil are conditions after treatments. We used "diseases and conditions" to describe the mixture in the main text. The main purpose of our study is to diagnose specific common retinal diseases and detect uncommon or rare diseases for referral. Using the mix of diseases and conditions is actually a novel strategy for classification, so that we could detect a wide spectrum of known diseases as well as unidentified diseases. We have attempted to be correct and clear in the revised manuscript.

Title: Add "and conditions" to become "Automatic detection of 39 fundus diseases and conditions in retinal photographs using deep neural networks".

Methods. Page 19, lines 380-397: "Some diseases shared similar characteristics and were not readily distinguishable according to fundus image only. For instance, hard exudate, sub-retinal hemorrhage, neovascularization, pigment epithelial detachment and macular atrophy can be found in wet AMD, PCV, choroidal neovascularization, macular atrophy, retinal angiomatous proliferation, and idiopathic macular telangiectasia. We clustered them together

and classified as the condition of maculopathy. Likewise, conditions causing optic disc swelling and elevation, such as papillitis, anterior ischemic optic neuropathy, papilledema, and pseudopapilloedema were classified as disc swelling and elevation. Other clustered conditions included optic nerve degeneration, posterior serous/exudative RD dragged disc, congenital abnormal disc, and fundus neoplasm. Some bigclasses were classified according to common features such as massive hard exudates, yellow-white spots/flecks, cotton-wool spots, vessel tortuosity, chorioretinal atrophy, coloboma, fibrosis and preretinal hemorrhage. Cases of rare diseases with these features could be included in such bigclasses, which were thus sufficiently large in number for deep feature pattern recognition. Post treatment conditions such as laser spots and silicon oil in eye were included to recognize images of patients after surgery. It also helped to identify cases after surgery that required follow-up."

Discussion. Page 14, lines 270-273: "First, categorization based on common retinal diseases and fundus features enables the detection of a wide spectrum of diseases, conditions and unidentified diseases. There are common features in different kinds of fundus diseases, including known or unknown rare diseases which can be detected by the DLP and treated as referable cases."

-- The external test sets were all DR, thus claiming that the performance on external datasets is perhaps overoptimistic

Response: We have test the DLP with datasets of other diseases including REFUGE (glaucoma) and PALM (pathological myopia). Results have been added to the revised manuscript.

Page 8, lines 152-174: "After testing with hetero-ethnic datasets in China, the generalization capabilities of our DLP to detect different diseases and conditions were evaluated with 4 public test datasets in the single disease setting, messidor-2, IDRID, PALM, and REFUGE (Supplementary Table 8). For detecting referable DR, we achieved a F1 score of 0.944, sensitivity of 0.906, specificity of 0.996 and AUC of 0.9861 in messidor-2. Performance was weaker in IDRID with F1 score of 0.875, sensitivity of 0.824, specificity of 0.902 and AUC of 0.9431. We reviewed the misjudged cases and found presence of stains on dirty lens looking like hemorrhage spots as the main cause of false positive results. For pathological myopia, higher performance in PALM was achieved with a F1 score of 0.974, sensitivity of 0.958, specificity of 0.988 and AUC of 0.9931. Performance was moderate for optic nerve degeneration (possible glaucoma) with a F1 score of 0.651 (0.674), sensitivity of 0.850 (0.813), specificity of 0.915 (0.933) and AUC of 0.9397 when compared to the top 12 contestant teams in the REFUGE Challenge 28. Labels of all images in the FEFUGE dataset were initially confirmed by multiple examinations including intraocular pressure (IOP), optical coherence tomography (OCT) and visual field. In early-stage glaucoma, almost no noticeable change could be detected by fundus though OCT could show retinal nerve fiber layer thinning. These cases were missed by the DLP which was developed based on fundus images only. Without specific optimization to the datasets of single diseases for competitions, the overall performance of our multi-label DLP with single disease datasets in different distribution was acceptable. These indicate good generalization capabilities in our DLP for detecting fundus diseases in heterogeneous images."

-- There is expected to be a lot of overlap between the categories. It is not clear how this overlap was dealt with.

Response: Yes, this is another limitation in our study. The dimensions of classes were not orthogonal to each other (an orthogonality example is color, size, and weight). There were overlaps between categories such as DR and Hard exudates. We have attempted multi-label classification that based on Binary Relevance to deal with these overlaps. We have added this limitation to the Discussion:

Page 17, lines 339-342: "The third limitation was existence of overlaps between classes such as referable DR and Hard exudates. Multi-label classification based on Binary Relevance was used to tackle both label overlaps and class overlaps. In order to eliminate class overlaps, some of these 39 classes should be further decomposed."

One way to tackle this issue is to decompose the current classes. For example, besides the DR international standard classification (grading 0-4), DR should be classified according to the type of lesions (hemorrhages, exudates, cotton wool spots). The class of Hard exudates also should be decomposed as exudates related to DR (This part coincides with the hard exudation subclass in DR) or not. But doing so will greatly increase the complexity of the system, we did not achieve this in this study, however we will take it as a research topic in the future.

-- Given that the architecture is not particularly novel, the performance of the algorithm seems extraordinarily high, given the current state of the art. Can you comment on what you believe has led to such high performance?

Response: Thanks for a very important comment which was also raised by #1 reviewer. We agree that we worked on a conventional rather than novel approach, though we have used dynamic data-resampling and weighted cross entropy loss function simultaneously to cope with the bias prediction in class imbalance situations, as well as used both CAM and DeepShap heatmaps to provide a certain degree of explainability. Our methodology was not very different from reported studies but we have attained better performance in AUC. But we think AUC can be over optimistic and unreliable in situations with huge imbalance ratios. F_1 should be more suitable for evaluating the performance of the DLP, but we kept providing AUC results for comparison. There should be two main reasons for our higher AUCs than those reported by others, different distribution of datasets and discarding of unclassifiable images and images with uncertain features in the labelling procedure. We have given explanation in the text with additional references.

Page 14, lines 278-293: "We obtained higher AUCs, mostly greater than 0.996, in both primary tests and multihospital tests in multi-label setting (Table 2) when compared with those attained from the public datasets or with reported studies for detecting single diseases, with reported AUC ranging 0.889-0.983 for referable DR¹⁹, 0.940 for large drusen²², 0.986 for DR³⁸ (Krause J. et al, *Ophthalmology* 2018). and 0.986 for glaucomatous optic neuropathy³⁹ (Li Z. et al, *Ophthalmology* 2018). Our apparently better performance was likely due to the extreme imbalance in the distribution of our dataset. We used multi-label setting, which included 30 bigclasses. Labels were very sparse and most samples belonged only to one label. The class imbalance ratios ranged from 0.7 to 4028.7 in our datasets. Consequently, ROC and AUC can be over optimistic and even unreliable in these situations^{40,41} (#40. Japkowicz N. and Holte R. 2000; #41. Saito T. and Rehmsmeier M. 2015). High AUC results were also reported in a study using multi-label setting with high imbalance ratios⁴² (Zhu H. et al. 2020). Therefore, F_1 scores should

be more suitable for evaluation of performance of the algorithms for multi-label setting with extreme imbalance ratios. Another possible explanation is that during the labeling procedure, some uncategorized images were discarded because of poor image quality or uncertain features as determined by the retina expert panel⁴³(Son J. et al.2020). The final results of AUC could thus be high."

-- *The github code suggest the use of an optic disk localizer based on the REFUGE challenge -was this used in this work?*

Response: Yes, the dataset used in optic disc detection contained both public datasets including REFUGE and IDRID and private dataset. We have added this information in the revised manuscript: "**Supplementary methods (Optic disc segmentation dataset).**"

We used an instance segmentation method i.e. Mask-RCNN instead of a localization method such as center point or BBOX regression because the confidence value Mask-RCNN outputted was important and the mask images of optic disc could be readily obtained.

Addition: "**Supplementary methods (Convolutional neural networks)**".

-- *Inclusion/exclusion of images without consensus labels should be clarified*

Response: Images were firstly labeled by unspecialized ophthalmologists, then confirmed by senior retina specialists. We clarified the procedure in the revised Method.

Page 23, lines 466-475: "Images were initially labeled by unspecialized ophthalmologists, then confirmed by senior retina specialists. Images that were agreed by the specialists were applied for deep learning directly. Otherwise, images were transferred to a retina expert panel of 5 senior retina specialists for final decision. Besides, we have added referable labels (observation, routine, semi-urgent, urgent) to all categories of diseases and conditions. For "blur fundus", additional suggestion for "repeat photography" was given (Supplementary Table 1). Unclassifiable images judged by the retina experts were excluded. They either had poor image quality or uncertain features⁴³, or were rare diseases that did not belong to the 39 categories (Supplementary Figure 2)."

--*Can you confirm that all ~250,000 images were manually rated by 20 ophthalmologists? Or clarify that process?*

Response: Yes, all images for the development of our DLP were labelled manually by of 20 ophthalmologists (arranged in 10 groups), followed by final judgement from the retina experts when the readings of unspecialized ophthalmologists and senior retina specialists were not consensus. The labeling procedures have been elaborated in the revised Methods.

Page 22, lines 461-470: "To ensure the accuracy of classification by the deep learning algorithms, the images were labeled manually by 20 licensed ophthalmologists in China before algorithm training, validation and testing. The ophthalmologists were separated into 10 groups. Each group had a senior retina specialist with more than 7 years' clinical experience and an unspecialized ophthalmologist having trained for over 3 years. Images were initially labeled by unspecialized ophthalmologists, then confirmed by senior retina specialists. Images that were agreed by the specialists were applied for deep learning directly.

Otherwise, images were transferred to a retina expert panel of 5 senior retina specialists for final decision."

-- *The utility and accuracy of the heatmaps are unclear. There is not quantification of their ability to localize.*

Response: Heatmaps can help to debug neural networks, for example the heatmaps can help to discover the neural networks making correct decisions, however, based on wrong areas. To a certain extent heatmaps can reveal why a neural network makes its decision of an image, though this ability is limited and unclear.

We wanted to quantify the performance of heatmaps. Unfortunately, under the conditions of multiple diseases and features, the quantitative analysis of heatmaps is very difficult. We have made quantitative analysis only for bigclass 10 (optic nerve degeneration), which showed 100% (1054) of their corresponding heatmaps are focus on the optic disc areas that was highly consistent with expert domain knowledge. We have clarified in the revised Discussion.

Page 16, lines 312-321: "Though quantitative evaluation on performance of heatmaps was difficult when there were multiple diseases and features, such facilities were capable to show how the DLP makes decisions by explicit fundus features including haemorrhages, exudates, hyperaemia and pale disc. Therefore, clinicians were able to "see" the lesion areas from the DLP and verify whether the DLP has used "appropriate" features for diagnosis. We have made quantitative analysis for bigclass 10 (optic nerve degeneration) true positive samples, which showed 100% (1,054 images) of their corresponding heatmaps were focused on the optic disc areas that was highly consistent with expert domain knowledge."

Minor -- "bigclasses" is a unusual term

Response: We admit that "bigblasses" is an unusual term. It meant the first hierarchy of classification, "subclasses" was used to denote the second hierarchy of classification. We did not use the term "classes" for the first hierarchy of classification because it could confuse with the concept of classification. We have given a definition for clarification.

Page 7, lines 114-115: "We termed major diseases or conditions as 'bigclass' for convenient classification and statistical analysis."

-- *not sure what "The sum of false negative (FN) and true positive (TP) represents the samples of corresponding class." means -can you clarify?*

Response: Sorry for the confusion. Correct description has been added.

Page 6, lines 104-105: "Positive samples of a class were obtained by summing up its false negatives (FN) and true positives (TP) accordingly."

Reviewer #3 (Remarks to the Author):

This is a major study reporting on the development and validation of an AI system to autonomously detect a range of retinal conditions in fundus photographs. The authors amassed a very extensive database of retinal images that was labelled manually for 39 classes. Testing

and validation was carried out in separate data sets, as well as external validation in independent datasets from the same region as well as public data sets. Furthermore, the system was tested in a clinical tele-reading setting, where it showed comparable performance to retina specialists.

The paper is the result of a huge interdisciplinary effort and at times challenging to navigate. For instance, when reading the paper, it is not entirely clear what “bigclass” is referring to, although this is important for understanding.

Response: As responded to 2# reviewer, we admit that “bigclasses” is an unusual term. It meant the first hierarchy of classification, “subclasses” was used to denote the second hierarchy of classification. We did not use the term “classes” for the first hierarchy of classification because it could confuse with the concept of classification. We have given a definition for clarification:

Page 7, lines 114-115: “We termed major diseases or conditions as ‘bigclass’ for convenient classification and statistical analysis.”

From a clinical standpoint I may offer a few major comments:

1) The 39 classes are a heterogeneous group of diagnoses (e.g. DR), signs (e.g. preretinal hemorrhage, disc swelling) and even postoperative findings (e.g. silicone oil in the eye). This appears as a collection of findings that were present in a certain dataset, and not as list of conditions that one would screen for in a medically oriented screening setting.

Response: Thanks for a very important comment. This is also a concern of Reviewer 2. We agree that the 39 classes are a mix of diseases, signs, and postoperative findings. However, we believed that this setting would be suitable for more application scenarios, and would not affect the screening efficacy for community screening. For example, in the tele-reading application, patients with fundus surgery history elsewhere were followed up in the primary hospitals, which could also be detected as conditions after surgery. The mixture of diseases and conditions is our novel strategy for classification, so that we can detect a wide spectrum of known diseases and unidentified diseases as many as possible. We have given clarifications in the revised manuscript.

Page 19, lines 380-397: “Some diseases shared similar characteristics and were not readily distinguishable according to fundus image only. For instance, hard exudate, sub-retinal hemorrhage, neovascularization, pigment epithelial detachment and macular atrophy can be found in wet AMD, PCV, choroidal neovascularization, macular atrophy, retinal angiomatous proliferation, and idiopathic macular telangiectasia. We clustered them together and classified as the condition of maculopathy. Likewise, conditions causing optic disc swelling and elevation, such as papillitis, anterior ischemic optic neuropathy, papilledema, and pseudopapilloedema were classified as disc swelling and elevation. Other clustered conditions included optic nerve degeneration, posterior serous/exudative RD dragged disc, congenital abnormal disc, and fundus neoplasm. Some bigclasses were classified according to common features such as massive hard exudates, yellow-white spots/flecks, cotton-wool spots, vessel tortuosity, chorioretinal atrophy, coloboma, fibrosis and preretinal hemorrhage. Cases of rare diseases with these features could be included in such bigclasses, which were thus sufficiently large in

number for deep feature pattern recognition. Post treatment conditions such as laser spots and silicon oil in eye were included to recognize images of patients after surgery. It also helped to identify cases after surgery that required follow-up."

Page 14, lines 270-273: "First, categorization based on common retinal diseases and fundus features enables the detection of a wide spectrum of diseases, conditions and unidentified diseases. There are common features in different kinds of fundus diseases, including known or unknown rare diseases which can be detected by the DLP and treated as referable cases."

2) When compared to a benchmark paper in the field that had a similar goal, although using a different image modality (OCT, reference # 33 De Fauw et al Nature Medicine), what I am missing most in this current paper is a triage of referable conditions (urgent, routine, etc.). For instance, retinal detachment is much more urgent than myelinated nerve fiber layer (which in fact is a congenital condition without clinical relevance, but listed as referable here). The presented work only provides non-referable vs. referable classes.

Response: Thanks for another important comment. We agree that triage of referable conditions is important and have added in the revised Method.

Page 23, lines 470-473: "Besides, we have added referable labels (observation, routine, semi-urgent, urgent) to all categories of diseases and conditions. For "blur fundus", additional suggestion for "repeat photography" was given (Supplementary Table 1)."

3) The ground truth labels were provided by 20 ophthalmologists in 10 teams of each a retina specialist and an ophthalmologist in training. Did the authors study how reproducible these 10 teams worked compared to each other and along their task (inter-grader and intra-grader reproducibility)? I did not see these data presented.

Response: We have added the statistics of inter-grader comparison in the supplementary document: "**Supplementary Table 3**".

REVIEWER COMMENTS

Reviewer #1 (Remarks to the Author):

Line 460. The numbers here don't match those in Supplementary Figure 2. For example, I don't see 9,126 anywhere for the OH dataset in that figure.

Line 473. I suggest adding statistics on how many images were excluded after the labeling process due to the judgement of "Unclassifiable images" by the retinal experts. If any of the images were discarded by the initial groups of labelers before being seen by the retinal experts, I suggest documenting that number as well.

Figure 4. I suggest adding a control flow for discarding 105 images that could not be categorized by the retinal specialist, to make the tele-reading process clear.

My other questions/comments have been addressed to my satisfaction.

Reviewer #2 (Remarks to the Author):

thank you for addressing the reviewer comments

Reviewer #3 (Remarks to the Author):

The authors have provided a thorough revision of the manuscript addressing all comments appropriately.

REVIEWER COMMENTS

Reviewer #1 (Remarks to the Author):

Line 460. The numbers here don't match those in Supplementary Figure 2. For example, I don't see 9,126 anywhere for the OH dataset in that figure.

Response: We are sorry for the unclear illumination. We have added the flow of image quality control procedure and showed the corresponding numbers of excluded images by the automatic quality control algorithm when the Image quality score was lower than 80 in Supplementary Figure 2.

Line 473. I suggest adding statistics on how many images were excluded after the labeling process due to the judgement of "Unclassifiable images" by the retinal experts. If any of the images were discarded by the initial groups of labelers before being seen by the retinal experts, I suggest documenting that number as well.

Response: Many thanks for the suggestion! We have included a data summary of the

excluded images by the ophthalmologists. We have added a supplementary table, the new Supplementary Table 3, which summarized the unclassifiable images. Description has also been added in the main text. (Page 5, lines 77-79 and page23, lines 477-479)

Figure 4. I suggest adding a control flow for discarding 105 images that could not be categorized by the retinal specialist, to make the tele-reading process clear.

Response: Thank you for the important suggestion! We have added the flow for discarding the uncategorized images in Figure 4. (Page 32, lines 726-730)

My other questions/comments have been addressed to my satisfaction.

Response: Thank you very much for giving us comments and suggestions to revise and improve our manuscript.

REVIEWERS' COMMENTS

Reviewer #1 (Remarks to the Author):

My comments have been addressed to my satisfaction.

REVIEWERS' COMMENTS

Reviewer #1 (Remarks to the Author):

My comments have been addressed to my satisfaction.

Response: Thank you very much for giving us comments and suggestions to revise and improve our manuscript.